# RSAM: Learning on manifolds with Riemannian Sharpness-aware Minimization

## Abstract

Nowadays, understanding the geometry of the loss landscape shows promise in enhancing a model's generalization ability. In this work, we draw upon prior works that apply geometric principles to optimization and present a novel approach to improve robustness and generalization ability for constrained optimization problems. Indeed, this paper aims to generalize the Sharpness-Aware Minimization (SAM) optimizer to Riemannian manifolds. In doing so, we first extend the concept of sharpness and introduce a novel notion of sharpness on manifolds. To support this notion of sharpness, we present a theoretical analysis characterizing generalization capabilities with respect to manifold sharpness, which demonstrates a tighter bound on the generalization gap, a result not known before. Motivated by this analysis, we introduce our algorithm, Riemannian Sharpness-Aware Minimization (RSAM). To demonstrate RSAM's ability to enhance generalization ability, we evaluate and contrast our algorithm on a broad set of problems, such as image classification and contrastive learning across different datasets, including CIFAR100, CIFAR10, and FGVCAircraft. Our code is publicly available here.

## 1 Introduction

One of the challenges in deep learning is the overfitting issue, which is attributed to high-dimensional and non-convex loss functions, which leads to complicated loss landscapes with multiple local minima. Hence, it is crucial to understand the geometry of the loss landscape to train robust models and improve generalization ability. Regarding this issue, flat minimizes, which seek for regions with low sharpness, have been known to be among the most effective approaches for robustness (Keskar et al., 2016; Kaddour et al., 2022b; Li et al., 2022). Indeed, Sharpness-Aware Minimization (SAM), as introduced by Foret et al. (2021b), stands out as a notable method by simultaneously minimizing the loss function and the worst-case loss within a neighborhood of the current model. Nowadays, SAM has proven to be versatile across a diverse array of tasks such as meta-learning (Abbas et al., 2022), federated learning (Qu et al., 2022), vision models (Chen et al., 2021), or language models (Bahri et al., 2022).

Another challenge within deep learning is that to preserve the model's robustness, it is often desirable to impose strict constraints on the parameters. Such constraints include the SPD constraints (Gao et al., 2020), orthogonality, and full rank (Xie et al., 2017; Roy et al., 2019; Wang et al., 2020). In such cases, the models are known to reside on some Riemannian manifolds, such as Grassmann manifolds, SPD manifolds, or Stiefel manifolds. Given the importance of understanding the geometry of the parameter space, especially when it is a differential manifold, various optimization techniques have been developed to learn on Riemannian manifolds (Bonnabel, 2013; Luenberger, 1972; Kasai et al., 2019; Sato et al., 2019; Zhang et al., 2017). Indeed, prior studies have demonstrated that taking into account this intrinsic geometry structure will remarkably improve the model's generalization ability (Roy et al., 2019; Absil et al., 2008a).

Understanding these two challenges in enhancing the model's robustness and generalization ability, we seek to generalize the SAM optimizer to Riemannian manifolds. In particular, we introduce a novel notion of sharpness on manifolds to study the intrinsic geometry of the parameter space and the loss landscape. This notion is backed by a comprehensive theoretical analysis that formulates generalization capacity in terms of neighborhood-wise training loss on the manifolds Indeed, our theorem establishes a tighter upper bound of $\mathcal{O}(\sqrt{d})$ compared to the existing bounds such as

$\mathcal{O}(\sqrt{k})$ from Foret et al. (2021b), in which $d$ is the dimensionality of the manifold, embedded in a higher dimensional Euclidean space with $k \gg d$ dimensions. Motivated by this theoretical analysis, we propose a Riemannian optimization technique called Riemannian Sharpness-Aware Minimization (RSAM). We show via an empirical study that RSAM improves the model's generalization ability across a range of different tasks such as supervised learning, self-supervised learning, and a diverse array of computer vision datasets (CIFAR100, CIFAR10, FGVCAircraft), as well as different models (ResNet34, ResNet50). Indeed, RSAM makes a notable improvement upon SAM and SupCon. We will also show in our ablation studies the ability of RSAM to seek flat regions on the loss landscape with the aid of Riemannian geometry. In short, our contributions are as follows:

- We introduce a novel notion of Sharpness on Riemannian manifolds, backed by a theoretical analysis establishing a tighter upper bound than the existing bounds.
- Motivated by the theory, we introduce RSAM and empirically study its efficacy across various settings. Our experiments show that RSAM outperforms current methods by notable margins.

## 2 RELATED WORKS

### 2.1 SHARPNESS AWARE MINIMIZATION

The Sharpness-Aware Minimization (SAM) technique, introduced by Foret et al. (2021a), has gained prominence due to its effectiveness and scalability compared to previous methods. SAM's versatility is evident across various tasks and domains, making it a powerful optimization approach. SAM has found applications in diverse areas such as meta-learning bi-level optimization (Abbas et al., 2022), federated learning (Qu et al., 2022), vision models (Chen et al., 2021), language models (Bahri et al., 2022), domain generalization (Cha et al., 2021), and multi-task learning (Phan et al., 2022).

Recent research has further enhanced SAM's capabilities by exploring its underlying geometry (Kwon et al., 2021; Kim et al., 2022), minimizing surrogate gaps (Zhuang et al., 2022), and speeding up training time (Du et al., 2022; Liu et al., 2022). Additionally, Kaddour et al. (2022a) empirically studied SAM's sharpness compared to SWA (Izmailov et al., 2018). In contrast, BSAM (Möllenhoff & Khan, 2023) demonstrated that SAM is an optimal Bayesian relaxation of standard Bayesian inference with a normal posterior. Moreover, Nguyen et al. (2023b) developed the sharpness concept for Bayesian Neural Networks. Finally, Nguyen et al. (2023a) generalized SAM by leveraging optimal transport-based distributional robustness with sharpness-aware minimization.

### 2.2 LEARNING ON MANIFOLD

Within the literature of machine learning, it is often desirable to impose constraints on a model's parameters of a model, such as orthogonality or full rank. In such cases, the search space is no longer an Euclidean space but a manifold. Studies in Riemannian geometry have indicated that considering the intrinsic geometry of the parameter space during training can yield better performance (Roy et al., 2019; Absil et al., 2008a). Indeed, the awareness of the intrinsic geometry can increase the likelihood of discovering desirable parameters and optimize training time by confining the search to a significantly lower-dimensional space.

As such, various classes of manifolds have been introduced across a spectrum of applications. For example, in the domain of metric learning, Roy et al. (2019) incorporated Stiefel manifolds to ensure that the learned parameters maintain orthogonality constraints. In the context of Gaussian mixture models, Gao et al. (2020) proposed a strategy involving learning on SPD manifolds to enforce SPD constraints. Furthermore, Grassmann manifolds have found applications in diverse domains, encompassing recommender systems (Dai et al., 2012; Boumal & Absil, 2015) or modeling affine subspaces within document-specific language models (Hall & Hofmann, 2000).

Various procedures to learn on Riemannian manifolds were proposed. One notable approach to enforce adherence to these manifold structures is the Riemannian gradient descent (RGD) algorithm (Luenberger, 1972). However, it is worth noting that RGD has computational limitations since it requires taking the whole dataset for each iteration. To address this issue, Bonnabel (2013) introduced the Riemannian stochastic gradient descent (RSGD) method, which reduces computational overhead and thus gains widespread adoption and is used on various manifolds such as SPD manifolds.

## 3 RSAM: RIEMANNIAN SHARPNESS-AWARE MINIMIZATION

### 3.1 FORMULATIONS AND NOTATIONS

This section presents the problem formulations and the notions used in our theory development. We consider a model $f_{\boldsymbol{\theta}} : X \to Y$ parameterized by parameter $\boldsymbol{\theta}$, where $X$ is the data space and $Y$ is the label space. Let $\mathcal{D}$ be the data/label distribution that generates data/label pair $(\mathbf{x}, \mathbf{y}) \sim \mathcal{D}$. Based on $\mathcal{D}$, we sample a specific training set $\mathcal{S} = \{(\mathbf{x}_1, \mathbf{y}_1), (\mathbf{x}_2, \mathbf{y}_2), \cdots, (\mathbf{x}_n, \mathbf{y}_n)\}$, where $(\mathbf{x}_1, \mathbf{y}_1), \ldots, (\mathbf{x}_n, \mathbf{y}_n) \overset{\text{iid}}{\sim} \mathcal{D}$.

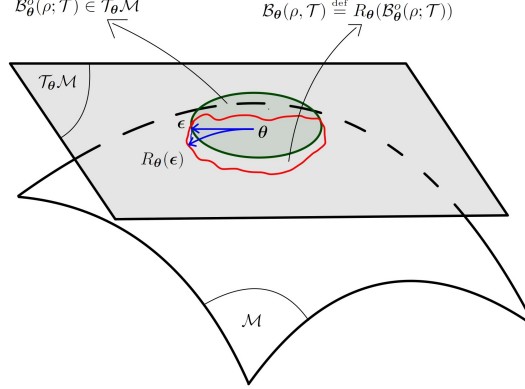

**Figure 1:** Neighborhoods on manifolds. $\mathcal{B}_{\boldsymbol{\theta}}(\rho; \mathcal{T})$ (green) is an $\rho$-ball on the tangent space $\mathcal{T}_{\boldsymbol{\theta}}\mathcal{M}$. The neighborhood $\mathcal{B}_{\boldsymbol{\theta}}(\rho; \mathcal{M})$ (red) of $\boldsymbol{\theta}$ on $\mathcal{M}$ is the retraction image of $\mathcal{B}_{\boldsymbol{\theta}}^o(\rho; \mathcal{T})$.

Given $(x, y) \sim \mathcal{D}$, we use the per-sample loss function $\ell(f_{\boldsymbol{\theta}}(\mathbf{x}), \mathbf{y})$ to quantify the loss suffered by the model $f_{\boldsymbol{\theta}}$ when predicting $(\mathbf{x}, \mathbf{y})$. At such, the empirical loss on the training set $\mathcal{S}$ is $\mathcal{L}_{\mathcal{S}}(\boldsymbol{\theta}) = \frac{1}{n} \sum_{i=1}^{n} \ell(f_{\boldsymbol{\theta}}(\mathbf{x}_i), \mathbf{y}_i)$, while the general loss on the data/label distribution $\mathcal{D}$ is $\mathcal{L}_{\mathcal{D}}(\boldsymbol{\theta}) = \mathbb{E}_{(\mathbf{x}, \mathbf{y}) \sim \mathcal{D}}[\ell(f_{\boldsymbol{\theta}}(\mathbf{x}), \mathbf{y})]$.

In this work, we assume that some constraints are imposed on the models (e.g., orthogonality, full rank, or SPD constraints), making the model parameters $\boldsymbol{\theta}$ lying in a *low-dimensional manifold* $\mathcal{M} \subset \mathbb{R}^k$ embedded in the *ambient vector space* $\mathbb{R}^k$. In other words, we assume that $\boldsymbol{\theta} \in \mathcal{M}$, in which $\mathcal{M}$ has dimensionality $d \ll k$. Bearing this in mind, we further develop Riemannian Sharpness-aware Minimization in the next section. For our theory development, we also introduce a few additional relevant notions. Indeed, for a point $\boldsymbol{\theta} \in \mathcal{M}$, we denote $\mathcal{T}_{\boldsymbol{\theta}}\mathcal{M}$ as the tangent space of $\mathcal{M}$. By convention, the tangent space $\mathcal{T}_{\boldsymbol{\theta}}\mathcal{M}$ uses the coordinate system with the current $\boldsymbol{\theta}$ as an origin hence $\boldsymbol{\epsilon} \in \mathcal{T}_{\boldsymbol{\theta}}\mathcal{M}$ specifies the offset from $\boldsymbol{\theta}$, thus further representing the vector $\boldsymbol{\theta} + \boldsymbol{\epsilon}$ on the ambient vector space $\mathbb{R}^k$. On the tangent space, we need to define the neighborhood (e.g., the $\rho$-ball) around $\boldsymbol{\theta}$ for presenting the concept of sharpness over the tangent space and the manifold. To serve our theory development, we define two equivalent $\rho$-balls w.r.t. the tangent and ambient space coordinate systems: $\mathcal{B}_{\boldsymbol{\theta}}^o(\rho; \mathcal{T}) = \{\boldsymbol{\epsilon} \in \mathcal{T}_{\boldsymbol{\theta}}\mathcal{M} : \|\boldsymbol{\epsilon}\|_2 \leq \rho\}$ (i.e., the offset ball) and $\mathcal{B}_{\boldsymbol{\theta}}^a(\rho; \mathcal{T}) = \boldsymbol{\theta} + \mathcal{B}_{\boldsymbol{\theta}}^o(\rho; \mathcal{T}) = \{\boldsymbol{\theta} + \boldsymbol{\epsilon} : \boldsymbol{\epsilon} \in \mathcal{T}_{\boldsymbol{\theta}}\mathcal{M} \text{ and } \|\boldsymbol{\epsilon}\|_2 \leq \rho\}$ (i.e., the absolute ball). Furthermore, we define the neighborhood of $\boldsymbol{\theta}$ on the manifold $\mathcal{M}$ as the retraction map of the one on the tangent space: $\mathcal{B}_{\boldsymbol{\theta}}(\rho; \mathcal{M}) = R_{\boldsymbol{\theta}}(\mathcal{B}_{\boldsymbol{\theta}}^o(\rho; \mathcal{T}))$, where $R_{\boldsymbol{\theta}}$ specifies the retraction operation at $\boldsymbol{\theta}$. For clarity, we refer to Figure 1 for an illustration of the neighborhood on Riemannian manifolds.

### 3.2 OUR THEORY DEVELOPMENT

In what follows, we present our theory development for RSAM. We consider the minimization problem in which the parameter space is an embedded submanifold $\mathcal{M} \in \mathbb{R}^k$:

$$\min_{\boldsymbol{\theta} \in \mathcal{M}} \mathcal{L}_{\mathcal{D}}(\boldsymbol{\theta})$$

Here, we can think of the condition $\boldsymbol{\theta} \in \mathcal{M}$ as a constraint to the optimization problem, such as orthogonality. There are two significant challenges regarding this constrained optimization problem. The first challenge is that $\mathcal{D}$ exists but is unknown and can only be realized through the training set $\mathcal{S}$. The second challenge, which is our main concern, is that $\boldsymbol{\theta}$ must remain within a manifold $\mathcal{M}$, i.e., satisfy the constraints. To tackle the second challenge, we present our main theorem about generalization ability on manifolds whose proof can be found in Appendix A.1.3.

**Theorem 1.** *For any small $\rho > 0$ and $\delta \in [0; 1]$, with a high probability $1 - \delta$ over training set $\mathcal{S}$ generated from a distribution $\mathcal{D}$, we have the following inequalities on the manifold $\mathcal{M}$:*

$$\mathcal{L}_\mathcal{D}(\boldsymbol{\theta}) \leq \max_{\boldsymbol{\theta}' \in \mathcal{B}_{\boldsymbol{\theta}}(\rho; \mathcal{M})} \mathcal{L}_\mathcal{S}(\boldsymbol{\theta}') + \mathcal{O}\left( C_\mathcal{M} \rho^2 + \sqrt{\frac{d + \log \frac{n}{\delta}}{n-1}} \right) \tag{1}$$

*in which $C_\mathcal{M}$ is a constant depends on $\mathcal{M}$.*

**Remark 1.** *The constant $C_\mathcal{M}$ depends on the structure manifold and may scale with dimensions for general manifolds. Additionally, for the retraction operations on the Stiefel manifold used in our practical algorithm and our experiments, the constant can be computed as $C_\mathcal{M} = 1 + \sqrt{2}/2$.*

This inequality expresses the generalization ability regarding neighborhood-wise training loss on the manifold instead of the whole ambient space as in Foret et al. (2021b). We note that $\mathcal{M}$ can have much smaller intrinsic dimensionality than the ambient space dimension, $d \ll k$. Therefore, this theorem gives us a tighter bound of $\mathcal{O}(d^{1/2})$ compared to $\mathcal{O}(k^{1/2})$ of SAM (Foret et al., 2021b). As a follow-up of this theoretical analysis, we also empirically demonstrate in Section 5.3.3 and Appendix 4c that our method did find the lower-sharpness region compared to prior works.

### 3.3 ALGORITHM

From the theorem above, we are thus motivated to find the local minimum in regions with small sharpness. Hence, motivated by the theoretical analysis, we propose to define the sharpness on manifolds as:

$$\max_{\boldsymbol{\theta}' \in \mathcal{B}_{\boldsymbol{\theta}}(\rho; \mathcal{M})} \mathcal{L}_\mathcal{S}(\boldsymbol{\theta}') - \mathcal{L}_\mathcal{S}(\boldsymbol{\theta})$$

Inspired by the term above, we propose to select the parameter values by solving the sharpness minimization problem:

$$\min_{\boldsymbol{\theta} \in \mathcal{M}} \mathcal{L}_\mathcal{S}^{RSAM}(\boldsymbol{\theta}), \text{ where } \mathcal{L}_\mathcal{S}^{RSAM}(\boldsymbol{\theta}) = \max_{\boldsymbol{\theta}' \in \mathcal{B}_{\boldsymbol{\theta}}(\rho; \mathcal{M})} \mathcal{L}_\mathcal{S}(\boldsymbol{\theta}')$$

Minimizing the equation above is equivalent to:

$$\min_{\boldsymbol{\theta} \in \mathcal{M}} \mathcal{L}_\mathcal{S}^{RSAM}(\boldsymbol{\theta}) = \min_{\boldsymbol{\theta} \in \mathcal{M}} \max_{\boldsymbol{\theta}' \in \mathcal{B}_{\boldsymbol{\theta}}(\rho; \mathcal{M})} \mathcal{L}_\mathcal{S}(\boldsymbol{\theta}') = \min_{\boldsymbol{\theta} \in \mathcal{M}} \max_{\boldsymbol{\theta}' \in R_{\boldsymbol{\theta}}(\mathcal{B}_{\boldsymbol{\theta}}^o(\rho; \mathcal{T}))} \mathcal{L}_\mathcal{S}(\boldsymbol{\theta}')$$

$$= \min_{\boldsymbol{\theta} \in \mathcal{M}} \left[ \mathcal{L}_\mathcal{S}(\boldsymbol{\theta}) + \left( \max_{\boldsymbol{\epsilon} \in \mathcal{B}_{\boldsymbol{\theta}}^o(\rho; \mathcal{T})} \mathcal{L}_\mathcal{S}(R_{\boldsymbol{\theta}}(\boldsymbol{\epsilon})) - \mathcal{L}_\mathcal{S}(\boldsymbol{\theta}) \right) \right]$$

$$= \min_{\boldsymbol{\theta} \in \mathcal{M}} \left[ \mathcal{L}_\mathcal{S}(\boldsymbol{\theta}) + \left( \max_{\boldsymbol{\epsilon} \in \mathcal{B}_{\boldsymbol{\theta}}^o(\rho; \mathcal{T})} \langle \text{grad}_{\boldsymbol{\theta}} \mathcal{L}_\mathcal{S}(\boldsymbol{\theta}), \boldsymbol{\epsilon} \rangle_{\boldsymbol{\theta}} + \mathcal{O}(||\boldsymbol{\epsilon}||_{\boldsymbol{\theta}}^2) \right) \right]$$

$$\approx \min_{\boldsymbol{\theta} \in \mathcal{M}} \left( \mathcal{L}_\mathcal{S}(\boldsymbol{\theta}) + \max_{\boldsymbol{\epsilon} \in \mathcal{B}_{\boldsymbol{\theta}}^o(\rho; \mathcal{T})} \langle \text{grad}_{\boldsymbol{\theta}} \mathcal{L}_\mathcal{S}(\boldsymbol{\theta}), \boldsymbol{\epsilon} \rangle_{\boldsymbol{\theta}} \right)$$

Where the third equality comes from the Taylor expansion on Riemannian manifold in Boumal (2023) and $\text{grad}_{\boldsymbol{\theta}} \mathcal{L}_\mathcal{S}(\boldsymbol{\theta})$ indicates the Riemannian gradient. Recall the definition that $\mathcal{B}_{\boldsymbol{\theta}}^o(\rho; \mathcal{M}) = \{\boldsymbol{\epsilon} \in \mathcal{T}_{\boldsymbol{\theta}} \mathcal{M} : ||\boldsymbol{\epsilon}||_2 \leq \rho\}$. We also define the Riemannian metric $\langle \boldsymbol{\epsilon}, \boldsymbol{\epsilon}' \rangle_{\boldsymbol{\theta}} = \boldsymbol{\epsilon}^\top \mathbf{D}_{\boldsymbol{\theta}} \boldsymbol{\epsilon}'$ for some matrix $\mathbf{D}_{\boldsymbol{\theta}}$ that reflects the local geometry at $\boldsymbol{\theta}$, the minimization problem is thus equivalent to:

$$\min_{\boldsymbol{\theta} \in \mathcal{M}} \max_{||\boldsymbol{\epsilon}||_2 \leq \rho, \boldsymbol{\epsilon} \in \mathcal{T}_{\boldsymbol{\theta}} \mathcal{M}} \left\langle \text{grad}_{\boldsymbol{\theta}}(\mathcal{L}_\mathcal{S}(\boldsymbol{\theta})), \boldsymbol{\epsilon} \right\rangle_{\boldsymbol{\theta}} \tag{2}$$

We first attempt to solve the inner maximization problem. Indeed, the problem has the following closed-form solution whose proof can be found in Appendix A.1.1.

**Proposition 1.** *Let $\text{grad}_\theta \mathcal{L}(\boldsymbol{\theta})^\top \mathbf{D}_{\boldsymbol{\theta}} = \mathbf{v}_{\boldsymbol{\theta}}^\top$ and $(\boldsymbol{u}_{\boldsymbol{\theta},j})$ be the system of orthonormal vectors of space formed by $\mathcal{T}_{\boldsymbol{\theta}} \mathcal{M}$. The closed-form solution to the maximization problem in Eq. (2) is given by:*

$$\boldsymbol{\epsilon}^* = \rho \sum_j \frac{\text{grad}_\theta \mathcal{L}(\boldsymbol{\theta})^\top \mathbf{D}_{\boldsymbol{\theta}} \boldsymbol{u}_{\boldsymbol{\theta},j}}{\sqrt{\sum_j \left[ \text{grad}_\theta \mathcal{L}(\boldsymbol{\theta})^\top \mathbf{D}_{\boldsymbol{\theta}} \boldsymbol{u}_{\boldsymbol{\theta},j} \right]^2}} \boldsymbol{u}_{\boldsymbol{\theta},j}$$

However, this closed-form solution is impractical in a wide range of cases. Firstly, due to the nested loop computation, the complexity scales poorly with respect to the dimensionality of $\mathcal{M}$. Moreover, finding the set of orthogonal vectors $(\mathbf{u}_{\boldsymbol{\theta},j})$ for a general manifold is not always straightforward in practice. Thus, we propose a more practical yet effective algorithm that first aims to find the solution $\overline{\epsilon}$ to the following relaxed problem:

$$\max_{\|\boldsymbol{\epsilon}\|_2 \leq \rho} \text{grad}_{\boldsymbol{\theta}}(\mathcal{L}_{\mathcal{S}}(\boldsymbol{\theta}))^{\top} \mathbf{D}_{\boldsymbol{\theta}} \boldsymbol{\epsilon} \tag{3}$$

and then project the solution onto the tangent space $\mathcal{T}_{\boldsymbol{\theta}}\mathcal{M}$ to get $\boldsymbol{\epsilon}^* = \text{Proj}_{\boldsymbol{\theta}}(\overline{\epsilon})$, which gives us an approximated solution to the maximization problem. Indeed, Eq. (3) yields the following solution, whose proof can be found in Appendix A.1.1.

**Proposition 2.** *The solution to the maximization problem in Eq. (3) is given by:*

$$\overline{\epsilon} = \rho \frac{grad_{\boldsymbol{\theta}}(\mathcal{L}(\boldsymbol{\theta}))^{\top} \mathbf{D}_{\boldsymbol{\theta}}}{\|grad_{\boldsymbol{\theta}}(\mathcal{L}(\boldsymbol{\theta}))^{\top} \mathbf{D}_{\boldsymbol{\theta}}\|_2}$$

After finding $\overline{\epsilon}$, we project the solution onto the tangent space and derive the approximated solution $\boldsymbol{\epsilon}^* = \text{Proj}_{\boldsymbol{\theta}}(\overline{\epsilon})$ to the maximization problem in Eq. 2. We will use this approximated solution for RSAM throughout this work, showing that it remarkably improves generalization ability in practice. Moreover, we empirically demonstrate in Section 5.3.1 that compared to the previous exact computation, this approach is notably more efficient and yet remains the same performance. Also, this approximated approach is much more flexible and applicable to a broad category of manifolds since the computation does not involve the orthogonal vectors of the manifolds. One may also notice that we use a matrix $\mathbf{D}_{\boldsymbol{\theta}}$ that can be adapted to learn the local metric at $\boldsymbol{\theta}$. The choice of this matrix is flexible. It can be either $\mathbf{D}_{\boldsymbol{\theta}} = \text{diag}(|\boldsymbol{\theta}_1|, |\boldsymbol{\theta}_2|, \cdots, |\boldsymbol{\theta}_k|)$, or $\mathbf{D}_{\boldsymbol{\theta}} = \mathbf{I}$. In our empirical studies, we use the former and refer to Section 5.3.2 for comparisons between these choices. Then, we solve the outer minimization problem with Riemannian gradient descent. In short, we summarize our algorithm Riemannian Sharpness-Aware Minimization as per Algorithm 1.

---

**Algorithm 1** Riemannian Sharpness-aware Minimization (RSAM)

---

**Input** Riemannian manifold $\mathcal{M}$, training set $\mathcal{S} \doteq \cup_{i=1}^{n}\{(\mathbf{x}_i, \mathbf{y}_i)\}$. Loss function $\ell : \mathcal{W} \times \mathcal{X} \times \mathcal{Y} \mapsto \mathbb{R}^+$, batch size $b$, learning rate $\eta > 0$, neighborhood size $\rho > 0$.
**Output:** Model trained with SAM on manifolds
Initialize weight $\theta_0$ on the manifold $\mathcal{M}$, $t = 0$
**while** *not converge* **do**
  Sample mini batch $\mathcal{B} = \{(\mathbf{x}_1, \mathbf{y}_1), \cdots, (\mathbf{x}_b, \mathbf{y}_b)\}$
  Compute the batch Riemannian gradient $\text{grad}_{\boldsymbol{\theta}}\mathcal{L}_{\mathcal{B}}(\theta) = \text{Proj}_{\boldsymbol{\theta}}(\nabla\mathcal{L}_{\mathcal{B}}(\boldsymbol{\theta}))$
  Compute $\overline{\epsilon} = \rho \frac{(\text{grad}_{\boldsymbol{\theta}}\mathcal{L}_{\mathcal{B}}(\theta))^{\top} \mathbf{D}_{\boldsymbol{\theta}}}{\left\|(\text{grad}_{\boldsymbol{\theta}}\mathcal{L}_{\mathcal{B}}(\theta))^{\top} \mathbf{D}_{\boldsymbol{\theta}}\right\|_2}$, and $\boldsymbol{\epsilon}^* = \text{Proj}_{\boldsymbol{\theta}}(\overline{\epsilon})$
  *Ascend step:* Compute $\boldsymbol{\theta}^* = \mathcal{R}_{\boldsymbol{\theta}}(\boldsymbol{\epsilon}^*)$
  *Descend step:* Update $\boldsymbol{\theta}_{t+1} = \mathcal{R}_{\boldsymbol{\theta}_t}(-\eta\text{grad}_{\boldsymbol{\theta}}(\mathcal{L}_{\mathcal{B}}(\boldsymbol{\theta}^*)))$
**end while**

---

## 4 PRACTICAL METHODS FOR SELF-SUPERVISED AND SUPERVISED LEARNING

In this section, we discuss the practical applications of RSAM for two settings: *supervised learning* and *self-supervised learning*. We will demonstrate these applications empirically in the next section. Throughout this paper, we are specifically interested in the Stiefel manifolds, that is defined as:

**Definition 1** (The Stiefel Manifolds). *The set of $(n \times p)-$dimensional matrices, $p \leq n$, with orthogonal columns and Frobenius inner products forms a Riemannian manifold is called the Stiefel manifold $St(p, n)$*

$$St(p, n) \doteq \{\boldsymbol{X} \in \mathbb{R}^{n \times p} : \boldsymbol{X}^{\top}\boldsymbol{X} = \boldsymbol{I}_p\}$$

Absil et al. (2008b) proposed multiple retractions for Stiefel manifolds. For the sake of computational complexity, we suggest using the retraction: $R_\mathbf{X}(\varepsilon) = \text{qf}(\mathbf{X} + \varepsilon)$ in which $\text{qf}(\mathbf{A})$ denote the $\mathbf{Q}$ factor of the decomposition of $\mathbf{A} \in \mathbb{R}_*^{n \times p}$ as $\mathbf{A} = \mathbf{QR}$. The projection can also be derived as $\text{Proj}_\mathbf{X}(\mathbf{v}) = \mathbf{v} - \mathbf{X}\text{Sym}(\mathbf{X}^\top \mathbf{v})$ in which $\text{Sym}(\mathbf{A}) = \frac{1}{2}(\mathbf{A} + \mathbf{A}^\top)$. In this paper, we demonstrate the performance of the Stiefel manifold in two applications: imposing orthogonal convolutional filters in CNN and metric learning for supervised contrastive learning.

## 4.1 Metric Learning for Self-supervised Learning

In this section, we particularly consider the Supervised Contrastive (SupCon) methodology as proposed by Khosla et al. (2021). For a set of $N$ randomly sampled sample/label pairs, $\{\mathbf{x}_k, \mathbf{y}_k\}_{k=1 \cdots N}$, the corresponding batch used for training consists of $2N$ pairs, $\{\tilde{\mathbf{x}}_l, \tilde{\mathbf{y}}_l\}_{l=1, \cdots, 2N}$, where $\tilde{\mathbf{x}}_{2k}$ and $\tilde{\mathbf{x}}_{2k-1}$ are random augmentations of $\mathbf{x}_k$, and $\tilde{\mathbf{y}}_{2k-1} = \tilde{\mathbf{y}}_{2k} = \mathbf{y}_k$. We refer to a set of $N$ samples as a "batch" and the set of $2N$ samples as a "multiview batch". Within a multiviewed batch, let $i \in I = \{1, \cdots, 2N\}$ be the index of an arbitrary augmented sample, and let $j(i)$ be the index of the other augmented sample originating from the same source sample. The architecture of SupCon involves two components: **1)** The backbone Encoders, which we denote as $\text{Enc}(\cdot)$; and **2)** The projection head $\text{Proj}(\cdot)$, which is either a linear or

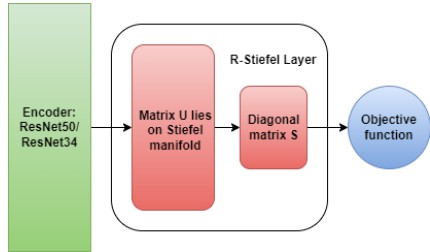

**Figure 2:** Metric learning with R-Stiefel layer. The projectional layer, which is typically a linear layer or an MLP, is replaced with the R-Stiefel layer consisting of a matrix $\mathbf{U} \in \text{St}(n, p)$ and a diagonal matrix $\mathbf{S}$.

fully-connected low-dimensional layer. For any $l$, we denote $\mathbf{z}_l = \text{Proj}(\text{Enc}(\tilde{\mathbf{x}}_l))$. We also define $A(i) = I/\{i\}$. As proposed by Khosla et al. (2021), $\mathbf{z}_l$'s are then trained with the SupCon objective:

$$\mathcal{L}_{out}^{sup} = \sum_{i \in I} \mathcal{L}_{out,i}^{sup} = \sum_{i \in I} \frac{-1}{|P(i)|} \log \frac{\exp(\mathbf{z}_i \cdot \mathbf{z}_p / \tau)}{\sum_{a \in A(i)} \exp(\mathbf{z}_i \cdot \mathbf{z}_a / \tau)}$$

$$= \mathcal{L}(\mathbf{z}_1 \cdots, \mathbf{z}_{2N}) = \mathcal{L}(\text{Proj}(f(\tilde{\mathbf{x}}_1)) \cdots, \text{Proj}(f(\tilde{\mathbf{x}}_{2N})))$$

Here, $P(i) = \{p \in A(i) : \tilde{\mathbf{y}}_p = \tilde{\mathbf{y}}_i\}$. Instead of using the Euclidean dot product, we replace it with the Mahalanobis distant $\langle \cdot, \cdot \rangle$ in which $\langle \mathbf{h}, \mathbf{h}' \rangle = \mathbf{h}^\top \mathbf{M} \mathbf{h}'$, and $\mathbf{M}$ is learnable. By doing so, $\mathbf{M}$ can be learned to take into account the local geometry of the parameter space, and the neighborhood becomes an adaptive ellipsoid instead of an open ball that treats every dimension identically. Singular Value Decomposition yields $\mathbf{M} = \mathbf{U}\mathbf{D}\mathbf{U}^\top = \mathbf{U}\mathbf{D}^{1/2}\mathbf{D}^{1/2}\mathbf{U}^\top$. Denote $\mathbf{S} = \mathbf{D}^{1/2}$, it follows that:

$$\langle \mathbf{h}, \mathbf{h}' \rangle = \mathbf{h}^\top \mathbf{M} \mathbf{h}' = (\mathbf{h}\mathbf{U}\mathbf{S})^\top \cdot (\mathbf{h}'\mathbf{U}\mathbf{S})$$

Thus, instead of optimizing $\mathcal{L}(\text{Proj}(\text{Enc}(\tilde{\mathbf{x}}_1)), \cdots, \text{Proj}(\text{Enc}(\tilde{\mathbf{x}}_{2N})))$, we will optimize $\mathcal{L}(\text{Proj}(\text{Enc}(\tilde{\mathbf{x}}_1))\mathbf{U}\mathbf{S}, \cdots, \text{Proj}(\text{Enc}(\tilde{\mathbf{x}}_{2N}))\mathbf{U}\mathbf{S})$ in which $\mathbf{U}$ is a rotational matrix on the Stiefel manifold, and $\mathbf{S}$ is a diagonal matrix. From now on, we will call the layer that multiplies with the matrix $\mathbf{U}\mathbf{S}$ an R-Stiefel layer. We refer to Figure 2 for illustration. Such modification can be done not only on the SupCon loss function but also on different loss functions involving distance calculations such as triplet loss (Roy et al., 2019). Since $\mathbf{U}$ is constrained to lies on the Stiefel manifold, we will optimize it with RSAM, and the rest of the parameters, including the backbone and the diagonal matrix $\mathbf{S}$, will be learned via traditional optimizers such as SAM or SGD.

## 4.2 Orthogonal Convolutional Neural Network

Orthogonality of convolutional filters has been proven to be useful for several purposes, such as alleviating gradient vanishing or exploding phenomenon (Xie et al., 2017), or decorrelating the filter banks so that they learn distinct features (Wang et al., 2020). For each $\ell$, let $\{\mathbf{W}_i\}_{i=1}^C$ be the set of convolutional kernels in $\ell$−th layer, in which $\mathbf{W}_i \in \mathbb{R}^{WHM}$. Previous works impose orthogonality by introducing orthogonal regularizers such as $\mathcal{L}_{\text{ortho}} = \frac{\lambda}{2} \sum_{i=1}^{D} \|\mathbf{W}_i^\top \mathbf{W} - \mathbf{I}\|_2^2$ (Xie et al., 2017), or a self-convolution regularization term of the kernels (Wang et al., 2020) to encourage orthogonality between the convolutional kernels. In this section, we propose eliminating those regularizers and

directly enforcing the kernels to be always orthogonal during training. Indeed, we flatten the kernels $\mathbf{W}_i$ into column vectors of shape $W \times H \times M$. Let $\mathbf{W}_\ell$ be the matrix with the columns formed by $\mathbf{W}'_i s$. The advantage of RSAM is that we can guarantee $\mathbf{W}_\ell$ always lies on the Stiefel manifold $\text{St}(W \times H \times M, C)$ during training, which means that:

$$\mathbf{W}_\ell^\top \mathbf{W}_\ell = \mathbf{I}_d$$

always holds throughout training, therefore guarantees orthonormality between the kernels on that specific layer $\ell$. By directly guaranteeing the parameters to reside within its true geometry, it is expected to improve the robustness of the learned model. In the next section, we will demonstrate that simply imposing orthogonality onto a single convolutional layer in the middle of the architecture by training with RSAM can improve generalization ability notably.

## 5 EXPERIMENTS

### 5.1 TRAINING DETAILS

To assess RSAM's efficacy, we experiment on various vision datasets (including CIFAR10, CI-FAR100, and FGVCAircraft) and different architectures (including ResNet50 and ResNet34). We conduct two sets of experiments: the standard supervised classification from scratch and supervised contrastive learning. All the experiments were trained for 500 epochs. The learning rates $\eta$ of SGD, SAM, and RSAM are set to $0.1$ with a cosine learning rate scheduler throughout the experiments. $\rho$ in SAM is set to $0.1$, and $\rho$ in RSAM is set to $0.5$. We trained our model with a batch size of 256 on CIFAR100 and CIFAR10. We specifically note that the FGVCAircraft dataset has a higher resolution, so we use a smaller batch size of 64 on this dataset for all methodologies.

### 5.2 EXPERIMENTAL RESULTS

**Supervised Learning.** We examine the classification accuracy with cross-entropy loss for the first set of experiments, optimizing by SGD, Sharpness Aware Minimization (SAM), and our algorithm RSAM. In this scenario, RSAM is used for imposing orthogonality of convolutional layers. Specifically, we imposed orthogonality on a single convolutional layer in the middle of the architecture in all settings. Table 1 shows that RSAM generalizes better than the baselines in this standard training setting, with an improvement of 1-3% compared to SGD and more than 1% higher than SAM.

| Method | CIFAR100 | | CIFAR10 | | FGVCAircraft | |
|---|---|---|---|---|---|---|
| | ResNet50 | ResNet34 | ResNet50 | ResNet34 | ResNet50 | ResNet34 |
| CE + SGD | 74.62 | 73.67 | 94.56 | 95.14 | 82.44 | 78.79 |
| CE + SAM | 75.04 | 75.05 | 95.39 | 95.52 | 83.01 | 80.56 |
| CE + RSAM | **77.78** | **76.36** | **96.32** | **96.10** | **84.68** | **83.12** |

**Table 1:** Top-1 classification accuracy for supervised learning settings. We compare cross-entropy training with SGD with momentum, SAM, and RSAM (Ours). RSAM is used to impose orthogonality convolutional of the filter banks.

**Self-Supervised Learning.** The second set of experiments has two stages. SupCon is trained with SGD with momentum, SAM, and RSAM in pretraining. Then, in the second stage, we conduct linear evaluation, that is, to freeze the parameters and train a linear classifier. We note that in the pretrained step, the projectional layer of SGD and SAM are linear layers, while RSAM's is the R-Stiefel layer as discussed in Section 4.1. Therefore, the applications of RSAM in this setting are two-fold: RSAM is used to impose orthogonality on the convolutional layers and used for the R-Stiefel during pretraining. As shown in Table 2, RSAM consistently outperforms the baselines. Furthermore, we note that on ResNet50, RSAM made a remarkable accuracy of 81.62% on CIFAR100, which outperforms 5% compared to SupCon on the same setting.

We also further note that RSAM involves additional manifold computations, which are expected to be slower. However, as we will show in Appendix. A.2.1, such difference in runtime is negligible and, therefore, worth the trade-off for better performance. Furthermore, we show in Appendix A.2.2

that RSAM has successfully found the low-sharpness region as suggested by our theoretical analysis.

| Method | CIFAR100 | | CIFAR10 | | FGVCAircraft | |
|---|---|---|---|---|---|---|
| | ResNet50 | ResNet34 | ResNet50 | ResNet34 | ResNet50 | ResNet34 |
| SupCon + SGD | 75.29 | 74.04 | 95.99 | 95.34 | 82.03 | 78.19 |
| SupCon + SAM | 76.73 | 76.91 | 96.31 | 96.07 | 82.84 | 81.73 |
| SupCon + RSAM | **81.62** | **80.51** | **96.86** | **96.65** | **84.73** | **84.52** |

**Table 2:** Top-1 classification accuracy for self-supervised learning settings with SupCon loss. SupCon is trained with SGD with momentum, SAM, and RSAM (Ours). RSAM is used for metric learning and imposing orthogonality on the convolutional operations.

## 5.3 ABLATION STUDIES

### 5.3.1 APPROXIMATED SOLUTION VS. CLOSED-FORM SOLUTION COMPARISION

In this section, we justify the usage of the approximated solution as per Eq. (2) instead of the exact solution as derived in Eq. (1) by contrasting the two approaches. It is noteworthy that the exact solution contains the orthogonal vectors $(\mathbf{u}_{\theta,j})$. For each matrix $\mathbf{X}$ on the Stiefel manifold, these vectors are given as the set of matrices $\{\mathbf{XS}\}$, where $\mathbf{S}$ is any $p$-by-$p$ matrices. However, for a general manifold, the computation of these orthogonal vectors is not always available. Thus, the approximated approach of RSAM gives us more flexibility and applies to other manifolds beyond the Stiefel manifolds. Moreover, the exact solution in Eq. (1) contains nested loops that sum over the orthogonal vectors. Therefore, it is expected to be $\mathcal{O}(p^2)$ asymptotically slower than RSAM. We refer to Table 3 for an empirical comparison, showing that both approaches have roughly the same accuracy. In contrast, the exact solution is about 1.75x times slower. Furthermore, when we imposed orthogonality on convolutional layers with more than 512 kernels, the exact solution failed to complete an epoch within a reasonable time. Thus, using the approximated solution as we did in RSAM is preferable.

| Method | CIFAR100 | | CIFAR10 | | FGVCAircraft | |
|---|---|---|---|---|---|---|
| | Accuracy | Runtime | Accuracy | Runtime | Accuracy | Runtime |
| RSAM (exact) | 75.21 | $78.5_{\pm 1.2}$ | **96.15** | $78.9_{\pm 1.3}$ | **83.31** | $154.5_{\pm 2.6}$ |
| RSAM (approx) | **76.36** | $\mathbf{52.6}_{\pm 1.7}$ | 96.10 | $\mathbf{51.1}_{\pm 1.9}$ | 83.12 | $\mathbf{133.0}_{\pm 1.4}$ |

**Table 3:** Comparision between the exact solution and approximated solution to Eq. (2) in terms of Top-1 classification accuracy and per-epoch wallclock runtime. The experiment is conducted on ResNet34.

### 5.3.2 CHOICES OF $\mathbf{D}_{\theta}$ COMPARISON

We recall that the procedure of RSAM in 1 involves the matrix $\mathbf{D}_{\theta}$, which serves to adjust the metric on the Euclidean tangent space. In Table 4, we compare the performance on the standard training settings for different choices of $\mathbf{D}_{\theta}$. As shown in the table, an adaptive choice for $\mathbf{D}_{\theta}$ does have an effect on the final performance, but this effect is negligible, and hence RSAM's performance is not highly dependent on the choice for this $\mathbf{D}_{\theta}$.

| Method | CIFAR100 | | CIFAR10 | |
|---|---|---|---|---|
| | ResNet34 | ResNet50 | ResNet34 | ResNet50 |
| $\mathbf{D}_{\theta} = \mathrm{diag}(\lvert\boldsymbol{\theta}_1\rvert, \lvert\boldsymbol{\theta}_2\rvert, \cdots, \lvert\boldsymbol{\theta}_k\rvert)$ | **76.36** | **77.78** | 96.10 | **96.32** |
| $\mathbf{D}_{\theta} = \mathbf{I}$ | 76.02 | 76.91 | **96.21** | 96.21 |

**Table 4:** Top-1 Accuracy on standard training settings for different choices of $\mathbf{D}_{\theta}$.

### 5.3.3 SAM VS. RSAM: BEHAVIORAL COMPARISON

In this ablation, we design a simple experiment on the dataset MNIST to show a particular case where RSAM is favorably robust. In particular, we train a simple PCA-style autoencoder that aims to find an orthogonal matrix $\mathbf{W}$ that encodes each input $\mathbf{x}$ into lower-dimensional $\mathbf{z} = \mathbf{xW}$, and then decodes as $\tilde{\mathbf{x}} = \mathbf{zW}^{\top}$. The encoded vector $\mathbf{z}$ is then used for the classification task. Therefore, the objective that we will minimize is the reconstruction loss with a classification loss act as a regularizer. Since $\mathbf{W}$ is constrained to be orthogonal, it lies on a Stiefel manifold. To enforce orthogonality with SAM, we need to add an *orthogonal regulazrizer* $\|\mathbf{W}^{\top}\mathbf{W} - \mathbf{I}_d\|_2^2$. Thus, our objective function is:

$$\mathcal{L}_{\mathcal{S}}(\mathbf{W}) = \frac{1}{n}\sum_{i=1}^{n}\|\mathbf{x}_i - \tilde{\mathbf{x}}_i\|_2^2 + \beta\mathrm{CrossEntropyLoss}(\mathbf{z}_i, \mathbf{y}_i) + \lambda\|\mathbf{W}^{\top}\mathbf{W} - \mathbf{I}_d\|_2^2$$

In this set of experiments, we set the batch size to 16, the learning rate to 0.1, $\beta = 0.1$, and $\rho = 0.3$. In Figure 3, we report **1)** the loss value over time, **2)** The sharpness of the loss function over time, and **3)** the values of the orthogonal regularize, which measures how orthogonal the parameters were. Indeed, in terms of loss function convergence, the smaller $\lambda$ is, the better SAM can keep up with RSAM because the orthogonal regularization has less impact on the final loss function. However, for smaller values $\lambda$, $\mathbf{W}$ fails to be orthogonal, demonstrating that SAM is remarkably sensitive to this orthogonal regularization. Hence, we emphasize that by directly enforcing $\mathbf{W}$ to be on the Stiefel manifold, RSAM eliminates this vulnerability, remarkably reducing the sharpness and leading to better loss convergence. In short, the ablation suggests that in certain scenarios, taking into account the intrinsic geometry of the parameters can notably enhance the model's robustness and performance, and we propose that RSAM has successfully done so.

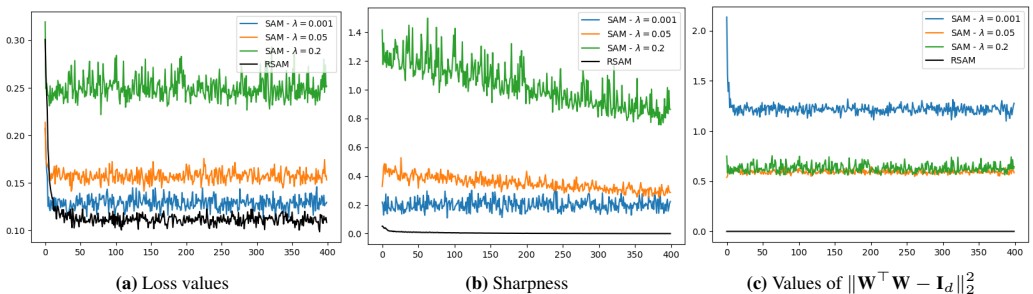

| (a) Loss values | (b) Sharpness | (c) Values of $\|\mathbf{W}^{\top}\mathbf{W} - \mathbf{I}_d\|_2^2$ |

**Figure 3:** Comparision between SAM with different $\lambda$'s and RSAM (black) over 50 epochs

## 6 CONCLUSION

In this work, we have generalized the SAM technique by introducing a novel notion of sharpness on Riemannian manifolds. On the theoretical side, we backed this notion with a theorem that characterizes the generalization ability in terms of neighborhood-wise training loss on the manifolds, which demonstrated a tighter bound compared to existing works. Motivated by this theoretical analysis, we propose RSAM, which considers the parameter space's intrinsic geometry and seeks regions with flat surfaces on Riemannian manifolds. On the experimental side, the effectiveness of RSAM is demonstrated on different tasks with various datasets and models, in which RSAM outperforms the comparative methodologies by a notable margin. Our code is publicly available here.

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

# A   APPENDIX

## A.1   ALL PROOFS

### A.1.1   PROOF OF PROPOSITION 1

Firstly, we restate the optimization problem:

$$\max_{\|\epsilon\|_2^2 \leq \rho^2, \epsilon \in \mathcal{T}_{\theta}\mathcal{M}} \big\langle \mathrm{grad}_{\theta}(\mathcal{L}_{\mathcal{S}}(\theta)), \epsilon \big\rangle_{\theta} \tag{4}$$

Let $\mathrm{grad}_{\theta}\mathcal{L}(\theta)^{\top}\mathbf{D}_{\theta} = \mathbf{v}_{\theta}^{\top}$ and $(\mathbf{u}_{\theta,j})$ be an orthonormal basis of $\mathcal{T}_{\theta}\mathcal{M}$. Then, it follows that:

$$\mathbf{u}_{\theta,i}^{\top}\mathbf{u}_{\theta,j} = \delta_{i,j}.$$

Under the assumption that the $\mathbf{u}_{\theta,j}$ form a basis in the tangent space at $\theta$, there exist $\beta_j$ such that

$$\epsilon = \sum_j \beta_j \mathbf{v}_{\theta,j}.$$

It deduces that

$$\epsilon^{\top}\epsilon = \Big[\sum_j \beta_j \mathbf{u}_{\theta,j}\Big]^{\top}\Big[\sum_j \beta_j \mathbf{u}_{\theta,j}\Big] = \sum_j \beta_j^2.$$

We have the Lagrangian objective being:

$$\mathbf{v}_{\theta}^{\top}\epsilon + \lambda\big[\epsilon^{\top}\epsilon - \rho^2\big] = \sum_j \beta_j \mathbf{v}_{\theta}^{\top}\mathbf{u}_{\theta,j} + \lambda\Big[\sum_j \beta_j^2 - \rho^2\Big].$$

Taking derivative with respect to $\lambda$ and $\beta_j$, we get the following system of equations:

$$\sum_j \beta_j^2 = \rho^2$$

$$\mathbf{v}_{\theta}^{\top}\mathbf{u}_{\theta,j} + 2\lambda\beta_j = 0.$$

Solving the second equation of the system yields:

$$\beta_j = -\frac{1}{2\lambda}\mathbf{v}_{\theta}^{\top}\mathbf{u}_{\theta,j}$$

Substituting into the first equation of the system, we get:

$$\frac{1}{4\lambda^2}\sum_j \big[\mathbf{v}_{\theta}^{\top}\mathbf{u}_{\theta,j}\big]^2 = \rho^2$$

$$\Rightarrow \frac{1}{2\lambda} = \pm\rho\Big\{\sum_j \big[\mathbf{v}_{\theta}^{\top}\mathbf{u}_{\theta,j}\big]^2\Big\}^{-\frac{1}{2}}$$

Then, the optimal solution to the maximization problem is given by:

$$\epsilon^* = \rho\sum_j \frac{\mathbf{v}_{\theta}^{\top}\mathbf{u}_{\theta,j}}{\sqrt{\sum_j \big[\mathbf{u}_{\theta}^{\top}\mathbf{u}_{\theta,j}\big]^2}}\mathbf{u}_{\theta,j}$$

$$= \rho\sum_j \frac{\mathrm{grad}_{\theta}\mathcal{L}(\theta)^{\top}\mathbf{D}_{\theta}\mathbf{u}_{\theta,j}}{\sqrt{\sum_j \big[\mathrm{grad}_{\theta}\mathcal{L}(\theta)^{\top}\mathbf{D}_{\theta}\mathbf{u}_{\theta,j}\big]^2}}\mathbf{u}_{\theta,j}$$

### A.1.2   PROOF OF PROPOSITION 2

Firstly, we restate the optimization problem:

$$\max_{\|\epsilon\|_2 \leq \rho} \mathrm{grad}_{\theta}(\mathcal{L}_{\mathcal{S}}(\theta))^{\top}\mathbf{D}_{\theta}\epsilon \tag{5}$$

We prove that the optimal solution of the problem in Eq. (5) occurs on the boundary. Suppose on the contrary that the optimal solution is $\epsilon^*$ satisfies $(\epsilon^*)^\top \epsilon^* = (\rho^*)^2 < \rho^2$. Then, $-\mathrm{grad}_{\boldsymbol{\theta}} \mathcal{L}(\boldsymbol{\theta})^\top \mathbf{D}_{\boldsymbol{\theta}} \epsilon^* \leq 0$, otherwise we may replace $\epsilon^*$ with $-\epsilon^*$ that still satisfies the constraint and arrive at a strictly smaller objective. However, if we replace $\epsilon^*$ with $\overline{\epsilon} = \epsilon^* \sqrt{\frac{\rho}{\rho^*}}$, it follows:

$$\overline{\epsilon}^\top \overline{\epsilon} = \rho^2$$

and arrive at a smaller objective since $\overline{\epsilon} > \epsilon^*$, which is a contradiction since we are assuming that $\epsilon^*$ is the optimal solution.

Therefore, the optimal solution $\epsilon^*$ occurs on the boundary. Thus, the problem is reduced to:

$$\text{maximize: } \mathrm{grad}_{\boldsymbol{\theta}}(\mathcal{L}(\boldsymbol{\theta}))^\top \mathbf{D}_{\boldsymbol{\theta}} \epsilon \tag{6a}$$

$$\text{subject to: } \epsilon^\top \epsilon - \rho^2 = 0 \tag{6b}$$

This problem has the Lagrangian:

$$L(\epsilon, \lambda) = -\mathrm{grad}_{\boldsymbol{\theta}}(\mathcal{L}(\boldsymbol{\theta}))^\top \mathbf{D}_{\boldsymbol{\theta}} \epsilon + \lambda(\epsilon^\top \epsilon - \rho^2)$$

The stationary point of $L$ satisfies $\frac{\partial L}{\partial \epsilon} = 0$ and $\frac{\partial L}{\partial \lambda} = 0$, which is equivalent to:

$$-\mathrm{grad}_{\boldsymbol{\theta}}(\mathcal{L}(\boldsymbol{\theta}))^\top \mathbf{D}_{\boldsymbol{\theta}} + 2\lambda\epsilon = 0$$

$$\epsilon^\top \epsilon = \rho^2$$

Thus, we have the system of equations:

$$\epsilon = \frac{1}{2\lambda} \mathrm{grad}_{\boldsymbol{\theta}}(\mathcal{L}(\boldsymbol{\theta}))^\top \mathbf{D}_{\boldsymbol{\theta}}$$

$$\epsilon^\top \epsilon = \rho^2$$

Substituting the first equation into the second one, we get:

$$\frac{1}{4\lambda^2} \mathbf{D}_{\boldsymbol{\theta}}^\top \mathrm{grad}_{\boldsymbol{\theta}}(\mathcal{L}(\boldsymbol{\theta})) \mathrm{grad}_{\boldsymbol{\theta}}(\mathcal{L}(\boldsymbol{\theta}))^\top \mathbf{D}_{\boldsymbol{\theta}} = \rho^2$$

Which follows that:

$$\frac{1}{2\lambda} = \frac{\rho}{\left((\mathrm{grad}_{\boldsymbol{\theta}}(\mathcal{L}(\boldsymbol{\theta}))^\top \mathbf{D}_{\boldsymbol{\theta}})^\top (\mathrm{grad}_{\boldsymbol{\theta}}(\mathcal{L}(\boldsymbol{\theta}))^\top \mathbf{D}_{\boldsymbol{\theta}})\right)^{\frac{1}{2}}} = \frac{\rho}{\|\mathrm{grad}_{\boldsymbol{\theta}}(\mathcal{L}(\boldsymbol{\theta}))^\top \mathbf{D}_{\boldsymbol{\theta}}\|_2}$$

Therefore, the maximization problem governs a closed-form solution:

$$\overline{\epsilon} = \rho \frac{\mathrm{grad}_{\boldsymbol{\theta}}(\mathcal{L}(\boldsymbol{\theta}))^\top \mathbf{D}_{\boldsymbol{\theta}}}{\|\mathrm{grad}_{\boldsymbol{\theta}}(\mathcal{L}(\boldsymbol{\theta}))^\top \mathbf{D}_{\boldsymbol{\theta}}\|_2}$$

### A.1.3 PROOF OF THEOREM 1

Firstly, we state a lemma that we will be using, whose proof can be found

**Lemma 1.** *(Boumal et al., 2018) There exists a constant $C_{\mathcal{M}} > 0$ such that for any $\boldsymbol{\theta} \in \mathcal{M}$ and $\epsilon \in \mathcal{T}_{\boldsymbol{\theta}}\mathcal{M}$, the following holds:*

$$\|\mathcal{R}_{\boldsymbol{\theta}}(\epsilon) - \boldsymbol{\theta} - \epsilon\|_F \leq C_{\mathcal{M}} \|\epsilon\|_F^2$$

**Remark:** *The constant $C_{\mathcal{M}}$ depends on the manifold structure. Indeed, for retractions on the Stiefel manifold, the constant is independent of $(d, k)$ and can be computed explicitly. Specifically, when using the QR factorization or the polar decomposition as the retraction, we have $C_{\mathcal{M}} = 1 + \sqrt{2}/2$.*

Now, we are ready to prove our main theorem. Indeed, we restate the theorem statement:

**Theorem 1.** *Assuming that the loss function $\mathcal{L}$ is $K-$Lipschitz. Then, for any small $\rho > 0$ and $\delta \in [0; 1]$, with a high probability $1 - \delta$ over training set $\mathcal{S}$ generated from a distribution $\mathcal{D}$, the following holds:*

$$\mathcal{L}_{\mathcal{D}}(\boldsymbol{\theta}) \leq \max_{\boldsymbol{\theta}' \in \mathcal{B}_{\boldsymbol{\theta}}(\rho; \mathcal{M})} \mathcal{L}_{\mathcal{S}}(\boldsymbol{\theta}') + \mathcal{O}\left( C_{\mathcal{M}}\rho^2 + \sqrt{\frac{d + \log \frac{n}{\delta}}{n - 1}} \right)$$

*Proof.* We assume that $\mathcal{M}$ is a compact manifold. Then, for every $\varepsilon > 0$, there exists a set $\{\boldsymbol{\theta}_i\}_{i=1}^J$ of predefined points on the manifold $\mathcal{M}$ that forms an $\varepsilon$-net of $\mathcal{M}$ with respect to the geodesic distance on $\mathcal{M}$. Indeed, for each $\boldsymbol{\theta} \in \mathcal{M}$, there exists $i$ such that $\boldsymbol{\theta}$ lies inside a neighborhood of $\boldsymbol{\theta}_i$ such that $d_{\mathcal{M}}(\boldsymbol{\theta}_i, \boldsymbol{\theta}) = d_i < \varepsilon$, in which $d_{\mathcal{M}}$ is the geodesic distance on manifold $\mathcal{M}$. Since the loss function $\mathcal{L}$ is $K$-Lipschitz, we have:

$$\left| \mathcal{L}_{\mathcal{D}}(\boldsymbol{\theta}) - \mathcal{L}_{\mathcal{D}}(\boldsymbol{\theta}_i) \right| \leq Kd_i = K\varepsilon.$$

For any $\boldsymbol{\theta}' \in \mathcal{B}_{\boldsymbol{\theta}}(\rho, \mathcal{M})$, we define the logarithm map $\log_{\boldsymbol{\theta}_i}(\boldsymbol{\theta}') = \widetilde{\boldsymbol{\theta}}_i'$, it follows that:

$$\rho - d_i \leq d_{\mathcal{M}}(\boldsymbol{\theta}', \boldsymbol{\theta}_i) = \|\widetilde{\boldsymbol{\theta}}_i' - \boldsymbol{\theta}_i\|_2 \leq d_i + \rho.$$

We also have:

$$\left| \mathcal{L}_{\mathcal{S}}(\widetilde{\boldsymbol{\theta}}') - \mathcal{L}_{\mathcal{S}}(\boldsymbol{\theta}') \right| = \left| \mathcal{L}_{\mathcal{S}}\left( \boldsymbol{\theta}_i + (\widetilde{\boldsymbol{\theta}}_i' - \boldsymbol{\theta}_i) \right) - \mathcal{L}_{\mathcal{S}}\left( R_{\boldsymbol{\theta}_i}(\widetilde{\boldsymbol{\theta}}_i' - \boldsymbol{\theta}_i) \right) \right|$$
$$\leq KC_{\mathcal{M}} \left\| \widetilde{\boldsymbol{\theta}}_i' - \boldsymbol{\theta}_i \right\|_F^2 = KC_{\mathcal{M}}(\varepsilon + \rho)^2.$$

and for any $\widetilde{\boldsymbol{\theta}}_i' \in \mathcal{B}_{\boldsymbol{\theta}_i}^a(\rho - \varepsilon, \mathcal{T})$ and $R_{\boldsymbol{\theta}_i}(\widetilde{\boldsymbol{\theta}}_i' - \boldsymbol{\theta}_i) = \boldsymbol{\theta}'$, we have

$$d_{\mathcal{M}}(\boldsymbol{\theta}', \boldsymbol{\theta}) \leq d_{\mathcal{M}}(\boldsymbol{\theta}', \boldsymbol{\theta}_i) + d_{\mathcal{M}}(\boldsymbol{\theta}_i, \boldsymbol{\theta}) \leq \varepsilon + \rho - \varepsilon = \rho. \tag{9}$$

Therefore, it follows that:

$$\max_{\widetilde{\boldsymbol{\theta}}_i' \in \mathcal{B}_{\boldsymbol{\theta}_i}^a(\rho - \varepsilon, \mathcal{T})} \mathcal{L}_{\mathcal{S}}(\widetilde{\boldsymbol{\theta}}_i') \leq \max_{\boldsymbol{\theta}' \in \mathcal{B}_{\boldsymbol{\theta}}(\rho, \mathcal{M})} \mathcal{L}_{\mathcal{S}}(\boldsymbol{\theta}') + KC_{\mathcal{M}}(\varepsilon + \rho)^2. \tag{10}$$

Now, for each $i$, we consider a collection of pairs of distributions $(Q_i, P_i)$ on the tangent space $\mathcal{T}_{\boldsymbol{\theta}_i}\mathcal{M}$ which takes $\boldsymbol{\theta}_i$ to be its origin, in which

$$P_i = Q_i = \mathcal{N}(0, r^2 \mathbf{I}_d)$$

here $P_i$ is a prior distribution and $Q_i$ is the posterior distribution. According to the union bound theorem, with probability at least $1 - \delta$ in which $\delta_i = \delta/J$, the following holds:

$$\mathbb{E}_{\boldsymbol{\theta} \sim Q_i}\left[ \mathcal{L}_D(\boldsymbol{\theta}) \right] \leq \mathbb{E}_{\boldsymbol{\theta} \sim Q_i} \mathcal{L}_{\mathcal{S}}(\boldsymbol{\theta}) + \sqrt{\frac{\mathsf{KL}(Q_i \| P_i) + \log \frac{n}{\delta_i}}{2n - 2}} = \mathbb{E}_{\boldsymbol{\theta} \sim Q_i} \mathcal{L}_{\mathcal{S}}(\boldsymbol{\theta}) + \sqrt{\frac{\log \frac{n}{\delta_i}}{2n - 2}} \tag{11}$$

Under the assumption that adding Gaussian perturbation on the weight space does not improve the test error, we have:

$$\mathcal{L}_{\mathcal{D}}(\boldsymbol{\theta}_i) \leq \mathbb{E}_{\boldsymbol{\theta} \sim Q_i} \mathcal{L}_{\mathcal{D}}(\boldsymbol{\theta}). \tag{12}$$

Also, for $\mathbf{z} \sim \mathcal{N}(0, r^2 \mathbf{I}_d)$ we have the following concentration inequality:

$$\|\mathbf{z}\|_2^2 \leq r^2 d \left( 1 + \sqrt{\frac{\ln(n)}{d}} \right)^2 \leq (\rho - \varepsilon)^2 \Leftrightarrow r \leq (\rho - \varepsilon) \frac{1}{\sqrt{d}\left( 1 + \sqrt{\frac{\ln(n)}{d}} \right)}.$$

with probability at least $1 - 1/\sqrt{n}$. Thus, if we choose $r = \frac{\rho - \varepsilon}{\sqrt{d}\left( 1 + \sqrt{\frac{\ln(n)}{d}} \right)}$, it follows that:

$$\mathbb{E}_{\boldsymbol{\theta} \sim Q_i} \mathcal{L}(\boldsymbol{\theta}) \leq \left( 1 - \frac{1}{\sqrt{n}} \right) \max_{\|\mathbf{z}\|_2 \leq \rho - \varepsilon, z \in \mathcal{T}_{\boldsymbol{\theta}_i}\mathcal{M}} \mathcal{L}_{\mathcal{S}}(\boldsymbol{\theta}_i + \mathbf{z}) + \frac{1}{\sqrt{n}}$$

Combining this result with the inequalities at 12, 11, 10, and 9, we derive that

$$\mathcal{L}_{\mathcal{D}}(\boldsymbol{\theta}) \leq \mathcal{L}_{\mathcal{D}}(\boldsymbol{\theta}_i) + K\varepsilon \leq \mathbb{E}_{\boldsymbol{\theta} \sim Q_i}\left[\mathcal{L}_{\mathcal{D}}(\boldsymbol{\theta})\right] + K\varepsilon + \sqrt{\frac{\ln(n) + \ln(J) + \ln(\frac{1}{\delta})}{2n - 2}}$$

$$\leq \max_{\|\mathbf{z}\|_2 \leq \rho - \varepsilon, z \in \mathcal{T}_{\boldsymbol{\theta}_i}\mathcal{M}} \mathcal{L}_{\mathcal{S}}(\boldsymbol{\theta}_i + z) + \frac{1}{\sqrt{n}} + K\varepsilon + \sqrt{\frac{\ln(n) + \ln(J) + \ln(\frac{1}{\delta})}{2n - 2}}$$

$$\leq \max_{\boldsymbol{\theta}' \in \mathcal{B}_{\boldsymbol{\theta}}(\rho, \mathcal{M})} \mathcal{L}_{\mathcal{S}}(\boldsymbol{\theta}') + KC_{\mathcal{M}}(\varepsilon + \rho)^2 + K\varepsilon + \sqrt{\frac{\mathcal{O}(\ln(J) + \ln(n/\delta))}{n - 1}}$$

Now, we are left to bound $\ln J$. Recall that $\mathcal{M}$ is a $d-$dimensional manifold covered within $J$ $\varepsilon$-balls. If we denote $R_j$ to be the $\varepsilon-$ball with the center $\boldsymbol{\theta}_i$, then $\mathrm{vol}(R_j) = \mathcal{O}(\varepsilon^d)$, implying $J = \mathcal{O}(\max_j \mathrm{diam}(\mathcal{M})^d / r_j^d)$, thus $\log J = \mathcal{O}(d)$. Plugging this into the last inequality, we derive that:

$$\mathcal{L}_D(\theta) \leq \max_{\boldsymbol{\theta}' \in \mathcal{B}_{\boldsymbol{\theta}}(\rho, \mathcal{M})} \mathcal{L}_{\mathcal{S}}(\boldsymbol{\theta}') + KC_{\mathcal{M}}(\varepsilon + \rho)^2 + K\varepsilon + \sqrt{\frac{\mathcal{O}(d + \ln(n/\delta))}{n - 1}}$$

which concludes our proof. □

## A.2 ADDITIONAL EXPERIMENTS

### A.2.1 PER-EPOCH RUNTIME

This ablation compares the single-epoch wallclock runtimes of SGD, SAM, and RSAM. Indeed, it is expected that SAM and RSAM take at least twice as long as SGD because both SAM and RSAM involve double backward-forward each iteration. We especially note that since RSAM involves additional computations on a manifold, it is expected that RSAM would take longer than SAM. As shown in Table 5, while RSAM improves the final performance, its runtime is only about 6% slower than SAM, therefore worth the tradeoff.

| Method | CIFAR100 | | CIFAR10 | | AirCraft | |
|---|---|---|---|---|---|---|
| | ResNet34 | ResNet50 | ResNet34 | ResNet50 | ResNet34 | ResNet50 |
| SGD | $21.5_{\pm 1.73}$ | $40.1_{\pm 2.96}$ | $21.4_{\pm 1.72}$ | $38.8_{\pm 3.05}$ | $57.6_{\pm 2.59}$ | $114.5_{\pm 4.77}$ |
| SAM | $49.1_{\pm 1.68}$ | $83.9_{\pm 2.79}$ | $48.8_{\pm 1.62}$ | $84.7_{\pm 2.68}$ | $125.3_{\pm 1.3}$ | $245.6_{\pm 4.30}$ |
| RSAM | $52.6_{\pm 1.66}$ | $88.7_{\pm 3.14}$ | $51.1_{\pm 1.90}$ | $88.3_{\pm 2.79}$ | $133.0_{\pm 1.4}$ | $259.2_{\pm 3.82}$ |

**Table 5:** Per-epoch wall-clock runtime in seconds.

### A.2.2 SHARPNESS EVOLUTION AND HESSIAN SPECTRAL

Throughout this work, we have designed RSAM to actively seek local minima in the regions within a manifold with both low loss value and low sharpness. In this section, to further verify whether RSAM found the low-sharpness region, we first contrast the spectral of the Hessian for ResNet34 trained on CIFAR100 for 400 steps with RSAM and SAM. Indeed, the model trained with RSAM has a lower maximum eigenvalue (10.86 of RSAM vs. 13.32 of SAM), and RSAM has a flatter eigenvalue distribution as shown in the Figure 4, therefore suggests that RSAM entered the lower-sharpness region on the loss landscape. Besides, we also report the sharpness evolutions over iterations as shown in Figure 4c. These results together indicate that RSAM successfully seeks points in lower-sharpness regions within the loss landscape.

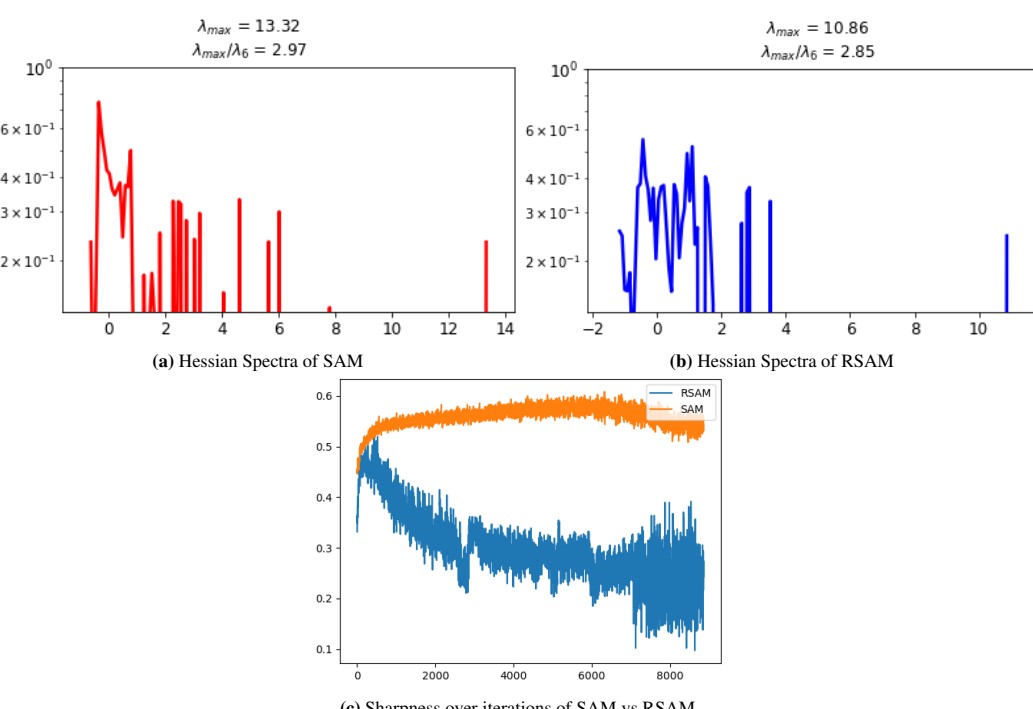

**(a)** Hessian Spectra of SAM

**(b)** Hessian Spectra of RSAM

**(c)** Sharpness over iterations of SAM vs RSAM.

**Figure 4:** The spectrum of the Hessian at the termination of the training phase with SAM vs. RSAM (above) and the evolution of sharpness over iterations (below). The results are reported on the CIFAR100 dataset with the SupCon loss function. $\rho$ in both methods is set to $0.1$

