{2}/2$. Hence, we arrive at the same upper bound as the first inequality in this case.*

The first inequality expresses the generalization ability regarding neighborhood-wise training loss on the tangent space instead of the whole ambient space as in Foret et al. (2021b). Since the tangent space locally reflects the geometry of the manifold around a sufficiently small neighborhood, it leads us to the second inequality, which captures the generalization ability within the manifold. We note that $\mathcal{M}$ can have much smaller intrinsic dimensionality than the ambient space dimension, $d \ll k$. Therefore, the theorem gives us a tighter bound of $\mathcal{O}(d^{1/2})$ compared to $\mathcal{O}(k^{1/2})$ of SAM (Foret et al., 2021b). As a follow-up of this theoretical analysis, we also empirically demonstrate in Section 5.3.3 and Appendix 4c that our method did find the lower-sharpness region compared to prior works.

### 3.3 ALGORITHM

From the theorems above, we are thus motivated to find the local minimum in regions with small sharpness. Hence, motivated by the theorems, we propose to define the sharpness on manifolds as:

$$\max_{\boldsymbol{\theta}' \in \mathcal{B}_{\boldsymbol{\theta}}(\rho;\mathcal{M})} \mathcal{L}_{\mathcal{S}}(\boldsymbol{\theta}') - \mathcal{L}_{\mathcal{S}}(\boldsymbol{\theta})$$

Inspired by the term above, we propose to select the parameter values by solving the sharpness minimization problem:

$$\min_{\boldsymbol{\theta} \in \mathcal{M}} \mathcal{L}_{\mathcal{S}}^{RSAM}(\boldsymbol{\theta}), \text{ where } \mathcal{L}_{\mathcal{S}}^{RSAM}(\boldsymbol{\theta}) = \max_{\boldsymbol{\theta}' \in \mathcal{B}_{\boldsymbol{\theta}}(\rho;\mathcal{M})} (\boldsymbol{\theta}')$$

Minimizing the equation above is equivalent to:

$$\min_{\boldsymbol{\theta} \in \mathcal{M}} \mathcal{L}_{\mathcal{S}}^{RSAM}(\boldsymbol{\theta}) = \min_{\boldsymbol{\theta} \in \mathcal{M}} \max_{\boldsymbol{\theta}' \in \mathcal{B}_{\boldsymbol{\theta}}(\rho;\mathcal{M})} (\boldsymbol{\theta}') = \min_{\boldsymbol{\theta} \in \mathcal{M}} \max_{\boldsymbol{\theta}' \in R_{\boldsymbol{\theta}}(\mathcal{B}_{\boldsymbol{\theta}}^o(\rho;\mathcal{T}))} \mathcal{L}_{\mathcal{S}}(\boldsymbol{\theta}')$$

$$= \min_{\boldsymbol{\theta} \in \mathcal{M}} \left[ \mathcal{L}_{\mathcal{S}}(\boldsymbol{\theta}) + \left( \max_{\boldsymbol{\epsilon} \in \mathcal{B}_{\boldsymbol{\theta}}^o(\rho;\mathcal{T})} \mathcal{L}_{\mathcal{S}}(R_{\boldsymbol{\theta}}(\boldsymbol{\epsilon})) - \mathcal{L}_{\mathcal{S}}(\boldsymbol{\theta}) \right) \right]$$

$$= \min_{\boldsymbol{\theta} \in \mathcal{M}} \left[ \mathcal{L}_{\mathcal{S}}(\boldsymbol{\theta}) + \left( \max_{\boldsymbol{\epsilon} \in \mathcal{B}_{\boldsymbol{\theta}}^o(\rho;\mathcal{T})} \langle \text{grad}_{\boldsymbol{\theta}} \mathcal{L}_{\mathcal{S}}(\boldsymbol{\theta}), \boldsymbol{\epsilon} \rangle_{\boldsymbol{\theta}} + \mathcal{O}(||\boldsymbol{\epsilon}||_{\boldsymbol{\theta}}^2) \right) \right]$$

$$\approx \min_{\boldsymbol{\theta} \in \mathcal{M}} \left( \mathcal{L}_{\mathcal{S}}(\boldsymbol{\theta}) + \max_{\boldsymbol{\epsilon} \in \mathcal{B}_{\boldsymbol{\theta}}^o(\rho;\mathcal{T})} \langle \text{grad}_{\boldsymbol{\theta}} \mathcal{L}_{\mathcal{S}}(\boldsymbol{\theta}), \boldsymbol{\epsilon} \rangle_{\boldsymbol{\theta}} \right)$$

Where the third equality comes from the Taylor expansion on Riemannian manifold in Boumal (2023) and $\text{grad}_{\boldsymbol{\theta}} \mathcal{L}_{\mathcal{S}}(\boldsymbol{\theta})$ indicates the Riemannian gradient. Recall the definition that $\mathcal{B}_{\boldsymbol{\theta}}^o(\rho;\mathcal{M}) = \{\boldsymbol{\epsilon} \in \mathcal{T}_{\boldsymbol{\theta}}\mathcal{M} : ||\boldsymbol{\epsilon}||_2 \leq \rho\}$. We also define the Riemannian metric $\langle \boldsymbol{\epsilon}, \boldsymbol{\epsilon}' \rangle_{\boldsymbol{\theta}} = \boldsymbol{\epsilon}^\top \mathbf{D}_{\boldsymbol{\theta}} \boldsymbol{\epsilon}'$ for some matrix $\mathbf{D}_{\boldsymbol{\theta}}$ that reflects the local geometry at $\boldsymbol{\theta}$, the minimization problem is thus equivalent to:

$$\min_{\boldsymbol{\theta} \in \mathcal{M}} \max_{||\boldsymbol{\epsilon}||_2 \leq \rho, \boldsymbol{\epsilon} \in \mathcal{T}_{\boldsymbol{\theta}}\mathcal{M}} \langle \text{grad}_{\boldsymbol{\theta}}(\mathcal{L}_{\mathcal{S}}(\boldsymbol{\theta})), \boldsymbol{\epsilon} \rangle_{\boldsymbol{\theta}} \tag{3}$$

We first attempt to solve the inner maximization problem. Indeed, the problem has the following closed-form solution whose proof can be found in Appendix A.1.1.

**Proposition 1.** *Let* $\mathrm{grad}_{\theta}\mathcal{L}(\theta)^{\top}\boldsymbol{D}_{\theta} = \boldsymbol{v}_{\theta}^{\top}$ *and* $(\boldsymbol{u}_{\theta,j})$ *be the system of orthonormal vectors of space formed by* $\mathcal{T}_{\theta}\mathcal{M}$. *The closed-form solution to the maximization problem in Eq. (3) is given by:*

$$\epsilon^{*} = \rho \sum_{j} \frac{\mathrm{grad}_{\theta}\mathcal{L}(\theta)^{\top}\boldsymbol{D}_{\theta}\boldsymbol{u}_{\theta,j}}{\sqrt{\sum_{j}\left[\mathrm{grad}_{\theta}\mathcal{L}(\theta)^{\top}\boldsymbol{D}_{\theta}\boldsymbol{u}_{\theta,j}\right]^{2}}}\boldsymbol{u}_{\theta,j}$$

However, this closed-form solution is impractical in a wide range of cases. Firstly, due to the nested loop computation, the complexity scales poorly with respect to the dimensionality of $\mathcal{M}$. Moreover, finding the set of orthogonal vectors $(\mathbf{u}_{\theta,j})$ for a general manifold is not always straightforward in practice. Thus, we propose a more practical yet effective algorithm that first aims to find the solution $\overline{\epsilon}$ to the following relaxed problem:

$$\max_{\|\epsilon\|_{2} \leq \rho} \mathrm{grad}_{\theta}(\mathcal{L}_{\mathcal{S}}(\theta))^{\top}\mathbf{D}_{\theta}\epsilon \qquad (4)$$

and then project the solution onto the tangent space $\mathcal{T}_{\theta}\mathcal{M}$ to get $\epsilon^{*} = \mathrm{Proj}_{\theta}(\overline{\epsilon})$, which gives us an approximated solution to the maximization problem. Indeed, Eq. (4) yields the following solution, whose proof can be found in Appendix A.1.1.

**Proposition 2.** *The solution to the maximization problem in Eq. (4) is given by*

$$\overline{\epsilon} = \rho \frac{grad_{\theta}(\mathcal{L}(\theta))^{\top}\boldsymbol{D}_{\theta}}{\|grad_{\theta}(\mathcal{L}(\theta))^{\top}\boldsymbol{D}_{\theta}\|_{2}}$$

After finding $\overline{\epsilon}$, we project the solution onto the tangent space and derive the approximated solution $\epsilon^{*} = \mathrm{Proj}_{\theta}(\overline{\epsilon})$ to the maximization problem in Eq. 3. We will use this approximated solution for RSAM throughout this work, showing that it remarkably improves generalization ability in practice. Moreover, we empirically demonstrate in Section 5.3.1 that compared to the previous exact computation, this approach is notably more efficient and yet remains the same performance. Also, this approximated approach is much more flexible and applicable to a broad category of manifolds since the computation does not involve the orthogonal vectors of the manifolds. One may also notice that we use a matrix $\mathbf{D}_{\theta}$ that can be adapted to learn the local metric at $\theta$. The choice of this matrix is flexible. It can be either $\mathbf{D}_{\theta} = \mathrm{diag}(|\theta_{1}|, |\theta_{2}|, \cdots, |\theta_{k}|)$, or $\mathbf{D}_{\theta} = \mathbf{I}$. In our empirical studies, we use the former and refer to Section 5.3.2 for comparisons between these choices. Then, we solve the outer minimization problem with Riemannian gradient descent. In short, we summarize our algorithm Riemannian Sharpness-Aware Minimization as per Algorithm 1.

---

**Algorithm 1** Riemannian Sharpness-aware Minimization (RSAM)

---

**Input** Riemannian manifold $\mathcal{M}$, training set $\mathcal{S} \doteq \cup_{i=1}^{n}\{(\mathbf{x}_{i}, \mathbf{y}_{i})\}$. Loss function $\ell : \mathcal{W} \times \mathcal{X} \times \mathcal{Y} \mapsto \mathbb{R}^{+}$, batch size $b$, learning rate $\eta > 0$, neighborhood size $\rho > 0$.
**Output:** Model trained with SAM on manifolds
Initialize weight $\theta_{0}$ on the manifold $\mathcal{M}$, $t = 0$
**while** *not converge* **do**
    Sample mini batch $\mathcal{B} = \{(\mathbf{x}_{1}, \mathbf{y}_{1}), \cdots, (\mathbf{x}_{b}, \mathbf{y}_{b})\}$
    Compute the batch Riemannian gradient $\mathrm{grad}_{\theta}\mathcal{L}_{\mathcal{B}}(\theta) = \mathrm{Proj}_{\theta}(\nabla\mathcal{L}_{\mathcal{B}}(\theta))$
    Compute $\overline{\epsilon} = \rho \frac{(\mathrm{grad}_{\theta}\mathcal{L}_{\mathcal{B}}(\theta))^{\top}\mathbf{D}_{\theta}}{\left\|(\mathrm{grad}_{\theta}\mathcal{L}_{\mathcal{B}}(\theta))^{\top}\mathbf{D}_{\theta}\right\|_{2}}$, and $\epsilon^{*} = \mathrm{Proj}_{\theta}(\overline{\epsilon})$
    *Ascend step:* Compute $\theta^{*} = R_{\theta}(\epsilon^{*})$
    *Descend step:* Update $\theta_{t+1} = R_{\theta_{t}}(-\eta\mathrm{grad}_{\theta}(\mathcal{L}_{\mathcal{B}}(\

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

Before going into the proof of our main theorems, we state an additional notation that we will be using throughout the proof. Indeed, for each $\boldsymbol{\theta} \in \mathcal{M}$, we define $A_{\boldsymbol{\theta}} \in \mathbb{R}^{k \times d}$ to be a column-orthogonal matrix whose columns form a basis of the tangent space $\mathcal{T}_{\boldsymbol{\theta}}\mathcal{M}$. We can construct $A_{\boldsymbol{\theta}}$ so that it is coordinate-wise smooth. To do so, notice that $\mathcal{M}$ is a $C^\infty$ embedded submanifold, so if we denote $U$ to be a neighborhood of $\boldsymbol{\theta}$ and a chart $\phi : (x^1, \cdots, x^k) : U \to \mathbb{R}^k$, we can define the ordered basis of $\mathcal{T}_{\boldsymbol{\theta}}\mathcal{M}$ as:

$$\forall i \in \{1, 2, \cdots, k\}, \forall f \in C^\infty(\mathcal{M}) : \frac{\partial}{\partial x^i}\Big|_{\boldsymbol{\theta}}(f) = \left(\frac{\partial}{\partial x^i}(f \circ \phi^{-1})\right)(\phi(\boldsymbol{\theta}))$$

We stacked the basis above into the columns of a matrix and then applied the Gram-Schmidt matrix; we got a smooth matrix $A_{\boldsymbol{\theta}}$ whose columns are orthogonal, that $A_{\boldsymbol{\theta}}^\top A_{\boldsymbol{\theta}} = \mathbf{I}_d$. Recall that by convention, the tangent space $\mathcal{T}_{\boldsymbol{\theta}}\mathcal{M}$ uses the coordinate system with the current $\boldsymbol{\theta}$ as the origin. Hence

$\epsilon \in \mathcal{T}_{\boldsymbol{\theta}}\mathcal{M}$ specifies the offset. Also, since the tangent space can locally reflect the geometry of the manifold, for each point $\boldsymbol{\theta} \in \mathcal{M}$, we only consider a sufficiently large and compact neighborhood $B$ of $\boldsymbol{\theta}$ on the tangent space, and that there is a collection of predefined regions $R_j$ such that $B \subset \cup_j R_j$ in which:

$$R_j = \left\{ \boldsymbol{\epsilon} \in \mathcal{T}_{\boldsymbol{\theta}}\mathcal{M} : (\boldsymbol{\epsilon} - \overline{\boldsymbol{\epsilon}}_j)^\top (A_{\boldsymbol{\theta}} A_{\boldsymbol{\theta}}^\top)^* (\boldsymbol{\epsilon} - \overline{\boldsymbol{\epsilon}}_j) \leq r_j^2 \right\},$$

Here, $\{\overline{\boldsymbol{\epsilon}}_j\}$ is a predefined set of points on $\mathcal{T}_{\boldsymbol{\theta}}\mathcal{M}$. We also define $r = \max_j r_j$, and additionally note that $A_{\boldsymbol{\theta}}^\top A_{\boldsymbol{\theta}} = \mathbf{I}_d$, for all $j = 1, 2, \ldots, J$, and $(A_{\boldsymbol{\theta}} A_{\boldsymbol{\theta}}^\top)^*$ denotes the Moore-Penrose inverse of $(A_{\boldsymbol{\theta}} A_{\boldsymbol{\theta}}^\top)$.

### A.1.4 PROOF OF FIRST INEQUALITY

*(The upper-bound on the tangent space $\mathcal{T}_{\boldsymbol{\theta}}\mathcal{M}$):* For any small $\rho > 0$ and $\delta \in [0; 1]$, with a high probability $1 - \delta$ over training set $\mathcal{S}$ generated from a distribution $\mathcal{D}$, the following holds:

$$\mathcal{L}_D(\boldsymbol{\theta}) \leq \max_{\boldsymbol{\theta}' \in \mathcal{B}_{\boldsymbol{\theta}}^a(\rho;\mathcal{T})} \mathcal{L}_S(\boldsymbol{\theta}') + \sqrt{\frac{\mathcal{O}(\frac{d}{\rho^2} + d + \log\frac{n}{\delta})}{n-1}}$$

*Proof.* Firstly, we recall the definition of the offset ball in the tangent space $\mathcal{B}_{\boldsymbol{\theta}}^o(\rho;\mathcal{M}) = \{\boldsymbol{\epsilon} \in \mathcal{T}_{\boldsymbol{\theta}}\mathcal{M} : \|\boldsymbol{\epsilon}\|_2 \leq \rho\}$, and the absolute ball that appears in the theorem statement $\mathcal{B}_{\boldsymbol{\theta}}^a(\rho;\mathcal{M}) = \boldsymbol{\theta} + \mathcal{B}_{\boldsymbol{\theta}}^o(\rho;\mathcal{M}) = \{\boldsymbol{\theta} + \boldsymbol{\epsilon} : \boldsymbol{\epsilon} \in \mathcal{T}_{\boldsymbol{\theta}}\mathcal{M} \text{ and } \|\boldsymbol{\epsilon}\|_2 \leq \rho\}$. Conventionally, the tangent space $\mathcal{T}_{\boldsymbol{\theta}}\mathcal{M}$ uses the coordinate system with $\boldsymbol{\theta}$ being the origin, so for $\boldsymbol{\theta}' = \boldsymbol{\theta} + \boldsymbol{\epsilon}$ on the tangent space, we can write $\mathcal{G}(\boldsymbol{\epsilon}) = \mathcal{L}(\boldsymbol{\theta}')$, which means that $\mathcal{L}(\boldsymbol{\theta}) = \mathcal{G}(0)$, and from now on we will analyze the $d-$dimensional Euclidean tangent space $\mathcal{T}_{\boldsymbol{\theta}}\mathcal{M}$. Accordingly, we will prove that:

$$\mathcal{G}_D(0) \leq \max_{\boldsymbol{\epsilon} \in \mathcal{T}_{\boldsymbol{\theta}}\mathcal{M}, \|\boldsymbol{\epsilon}\| \leq \rho} \mathcal{G}_S(\boldsymbol{\epsilon}) + \sqrt{\frac{\mathcal{O}(\frac{d}{\rho^2} + d + \log\frac{n}{\delta})}{n-1}}$$

Let $\delta > 0$ be a small positive constant. According to the PAC-Bayes generalization bound of McAllester (1999), for any prior distribution $P(\boldsymbol{\theta})$, with probability at least $1 - \delta$ over the choice of training set $S$, it holds that:

$$\mathbb{E}_{Q(\boldsymbol{\epsilon})}[\mathcal{G}_D(\boldsymbol{\epsilon})] \leq \mathbb{E}_{Q(\boldsymbol{\epsilon})}[\mathcal{G}_S(\boldsymbol{\epsilon})] + \sqrt{\frac{\mathsf{KL}(Q(\boldsymbol{\epsilon})\|P(\boldsymbol{\epsilon})) + \log\frac{n}{\delta}}{2n-2}}, \tag{10}$$

for any posterior distribution $Q(\boldsymbol{\epsilon})$ over the space of $\boldsymbol{\epsilon}$.

Here we consider $Q(\boldsymbol{\epsilon}) = \mathcal{N}(0, \rho^2 A_{\boldsymbol{\theta}} A_{\boldsymbol{\theta}}^\top)$. By imposing such posterior on $\boldsymbol{\epsilon}$, it means that we have $\boldsymbol{\epsilon} = A_{\boldsymbol{\theta}} h$ in which $h \sim \mathcal{N}(0, \rho^2 \mathbf{I}_d)$. To minimize the bound at Eq. 10, we want to choose the prior $P(\boldsymbol{\epsilon})$ that minimizes the KL term, and the prior should be independent of the training set $S$. Now we considered the predefined collection of points $\{\overline{\boldsymbol{\epsilon}}_j\}_{j=1}^J$ whose neighborhoods covers a sufficiently large neighborhood of $\boldsymbol{\theta}$ on the tangent space $\mathcal{T}_{\boldsymbol{\theta}}\mathcal{M}$, and a predefined set of $J$ prior distributions $\{P_j(\boldsymbol{\epsilon})\}_{j=1}^J$ in which $P_j(\boldsymbol{\epsilon}) = \mathcal{N}(\overline{\boldsymbol{\epsilon}}_j, \rho^2 A_{\boldsymbol{\theta}} A_{\boldsymbol{\theta}}^\top)$, and we select the closest distribution from this collection. According to the intersection of the training sets for which Eq. 10 holds, we can say that Eq. 10 holds for all $P_j(\boldsymbol{\epsilon})$ over the intersection. By the union bound theorem, the probability over the choice of the intersection is at least $1 - \sum_{j=1}^J \delta_j$. Letting $\delta_j = \delta/J$, we derive that for all $P_j(\boldsymbol{\epsilon})$, with probability at least $1 - \delta$ over the choice of the training set $S$, the following holds for all $Q(\boldsymbol{\epsilon})$ and $j = 1, \ldots, J$

$$\mathbb{E}_{Q(\boldsymbol{\epsilon})}[\mathcal{G}_D(\boldsymbol{\epsilon})] \leq \mathbb{E}_{Q(\boldsymbol{\epsilon})}[\mathcal{G}_S(\boldsymbol{\epsilon})] + \sqrt{\frac{\mathsf{KL}(Q(\boldsymbol{\epsilon})\|P_j(\boldsymbol{\epsilon})) + \log\frac{n}{\delta} + \log J}{2n-2}}$$

We choose the prior $P_j(\boldsymbol{\epsilon})$ as close to $Q(\boldsymbol{\epsilon})$. Indeed, the KL divergence term has the following form

$$\begin{aligned}
\mathsf{KL}(Q\|P_j) &= \frac{1}{2}\left[\mathsf{tr}\left((A_{\boldsymbol{\theta}} A_{\boldsymbol{\theta}}^\top)^*(A_{\boldsymbol{\theta}} A_{\boldsymbol{\theta}}^\top)\right) + \frac{1}{\rho^2}\overline{\boldsymbol{\epsilon}}_j^\top (A_{\boldsymbol{\theta}} A_{\boldsymbol{\theta}}^\top)^* \overline{\boldsymbol{\epsilon}}_j + \log\frac{|A_{\boldsymbol{\theta}} A_{\boldsymbol{\theta}}^\top|}{|A_{\boldsymbol{\theta}} A_{\boldsymbol{\theta}}^\top|} - k\right] \\
&= \frac{1}{2}\left[d - k + \frac{1}{\rho^2}\overline{\boldsymbol{\epsilon}}_j^\top (A_{\boldsymbol{\theta}} A_{\boldsymbol{\theta}}^\top)^* \overline{\boldsymbol{\epsilon}}_j\right]
\end{aligned}$$

Now, we choose $j^*$ such that $0 \in R_{j^*}$. According to our assumption, we have:

$$\bar{\boldsymbol{\epsilon}}_j^\top (A_{\boldsymbol{\theta}} A_{\boldsymbol{\theta}}^\top)^* \bar{\boldsymbol{\epsilon}}_j \leq r_j^2 \leq r^2$$

Therefore, we have the inequality:

$$\mathsf{KL}(Q \| P_j) \leq \frac{1}{2}(d - k + \frac{r^2}{\rho^2}) \leq \frac{d}{2} + \frac{r^2}{2\rho^2}$$

Plugging into the inequality 10, it follows that:

$$\mathbb{E}_{Q(\boldsymbol{\epsilon})}\big[\mathcal{G}_D(\boldsymbol{\epsilon})\big] \leq \mathbb{E}_{Q(\boldsymbol{\epsilon})}\big[\mathcal{G}_S(\boldsymbol{\epsilon})\big] + \sqrt{\frac{\frac{d}{2} + \frac{r^2}{2\rho^2} + \log \frac{n}{\delta} + \log J}{2n - 2}}$$

reverting from $\mathcal{G}$ back to $\mathcal{L}$, it means that:

$$\mathbb{E}_{\boldsymbol{\epsilon} \sim \mathcal{N}(0, \rho^2 A_{\boldsymbol{\theta}} A_{\boldsymbol{\theta}}^\top)}[\mathcal{L}_D(\boldsymbol{\theta} + \boldsymbol{\epsilon})] \leq \mathbb{E}_{\boldsymbol{\epsilon} \sim \mathcal{N}(0, \rho^2 A_{\boldsymbol{\theta}} A_{\boldsymbol{\theta}}^\top)}[\mathcal{L}_S(\boldsymbol{\theta} + \boldsymbol{\epsilon})] + \sqrt{\frac{\frac{d}{2} + \frac{r^2}{2\rho^2} + \log \frac{n}{\rho} + \log J}{2n - 2}}$$

Since $\boldsymbol{\epsilon} \sim \mathcal{N}(0, \rho^2 A_{\boldsymbol{\theta}} A_{\boldsymbol{\theta}}^\top)$, we can write $\boldsymbol{\epsilon} = A_{\boldsymbol{\theta}} \mathbf{z}$, in which $\mathbf{z} \sim \mathcal{N}(0, \rho^2 \mathbf{I}_d)$. Notice that $\|\boldsymbol{\epsilon}\|_2^2 = \boldsymbol{\epsilon}^\top \boldsymbol{\epsilon} = \mathbf{z}^\top A_{\boldsymbol{\theta}}^\top A_{\boldsymbol{\theta}} \mathbf{z} = \mathbf{z}^\top \mathbf{z} = \|\mathbf{z}\|_2^2$. We have the concentration inequality:

$$\mathbf{z} \sim \mathcal{N}(0, \rho^2 \mathbf{I}_d) \implies \|\mathbf{z}\|^2 \leq d\rho^2 \Big(1 + \sqrt{\frac{\log n}{d}}\Big)^2$$

with probability at least $1 - \frac{1}{\sqrt{n}}$. Thus, we have $\|\boldsymbol{\epsilon}\|^2 = \|\mathbf{z}\|_2^2 \leq d\rho^2(1 + \sqrt{\frac{\log n}{d}})^2$ with probability at least $1 - \frac{1}{\sqrt{n}}$. Denote $\gamma = \rho(\sqrt{d} + \sqrt{\log n})$, it follows that:

$$\mathbb{E}_{\boldsymbol{\epsilon} \sim \mathcal{N}(0, \rho^2 A_{\boldsymbol{\theta}_j} A_{\boldsymbol{\theta}_j}^\top)}[\mathcal{L}_S(\boldsymbol{\theta} + \boldsymbol{\epsilon})] \leq \Big(1 - \frac{1}{\sqrt{n}}\Big) \max_{\|\boldsymbol{\epsilon}\|_2^2 \leq \gamma^2} \mathcal{L}_S(\boldsymbol{\theta} + \boldsymbol{\epsilon}) + \frac{\mathcal{L}_{\max}}{\sqrt{n}}$$

$$\leq \max_{\|\boldsymbol{\epsilon}\|_2^2 \leq \gamma^2} \mathcal{L}_S(\boldsymbol{\theta} + \boldsymbol{\epsilon}) + \frac{\mathcal{L}_{\max}}{\sqrt{n}}$$

which implies

$$\mathbb{E}_{\boldsymbol{\epsilon} \sim \mathcal{N}(0, \rho^2 A_{\boldsymbol{\theta}_j} A_{\boldsymbol{\theta}_j}^\top)}[\mathcal{L}_D(\boldsymbol{\theta} + \boldsymbol{\epsilon})] \leq \max_{\|\boldsymbol{\epsilon}\|_2^2 \leq \gamma^2} \mathcal{L}_S(\boldsymbol{\theta} + \boldsymbol{\epsilon}) + \frac{\mathcal{L}_{\max}}{\sqrt{n}} + \sqrt{\frac{\frac{d}{2} + \frac{r^2}{2\rho^2} + \log \frac{n}{\delta} + \log J}{2n - 2}}$$

$$\leq \max_{\|\boldsymbol{\epsilon}\|_2^2 \leq \gamma^2} \mathcal{L}_S(\boldsymbol{\theta} + \boldsymbol{\epsilon}) + \frac{\mathcal{L}_{\max}}{\sqrt{n}}$$

$$+ \sqrt{\frac{\frac{d}{2} + \frac{r^2(\sqrt{d} + \sqrt{\log n})^2}{2\gamma^2} + \log \frac{n}{\delta} + \log J}{2n - 2}}$$

Now, we are left to bound $\log J$. Recall from our assumption that $R_j = \big\{\boldsymbol{\epsilon} \in \mathcal{T}_{\boldsymbol{\theta}}\mathcal{M} : (\boldsymbol{\epsilon} - \bar{\boldsymbol{\epsilon}}_j)^\top (A_{\boldsymbol{\theta}} A_{\boldsymbol{\theta}}^\top)^* (\boldsymbol{\epsilon} - \bar{\boldsymbol{\epsilon}}_j) \leq r_j^2\big\}$, and since the tangent space also has $d$ dimensions, so $\mathrm{vol}(R_j) = \mathcal{O}(r_j^d)$, which means $J = \mathcal{O}(\max_j \mathrm{diam}(\mathcal{M})^d / r_j^d)$, thus $\log J = \mathcal{O}(d)$. We derive that:

$$\mathbb{E}_{\boldsymbol{\epsilon} \sim \mathcal{N}(0, \rho^2 A_{\boldsymbol{\theta}_j} A_{\boldsymbol{\theta}_j}^\top)}[\mathcal{L}_D(\boldsymbol{\theta} + \boldsymbol{\epsilon})]$$

$$\leq \max_{\|\boldsymbol{\epsilon}\|_2^2 \leq \gamma^2, \boldsymbol{\epsilon} \in \mathcal{T}_{\boldsymbol{\theta}}\mathcal{M}} \mathcal{L}_S(\boldsymbol{\theta} + \boldsymbol{\epsilon}) + \frac{\mathcal{L}_{\max}}{\sqrt{n}} + \sqrt{\frac{\frac{d}{2} + \frac{r^2(\sqrt{d} + \sqrt{\log n})^2}{2\gamma^2} + \log \frac{n}{\delta} + \log J}{2n - 2}}$$

$$\leq \max_{\|\boldsymbol{\epsilon}\|_2^2 \leq \gamma^2, \boldsymbol{\epsilon} \in \mathcal{T}_{\boldsymbol{\theta}}\mathcal{M}} \mathcal{L}_S(\boldsymbol{\theta} + \boldsymbol{\epsilon}) + \sqrt{\frac{\frac{r^2(\sqrt{d} + \sqrt{\log n})^2}{2\gamma^2} + \log \frac{n}{\delta} + O(d)}{2n - 2}}$$

Under the assumption that adding Gaussian perturbation on the weight space does not improve the test error, we have:

$$\mathcal{L}_D(\boldsymbol{\theta}) \leq \mathbb{E}_{\boldsymbol{\epsilon} \sim \mathcal{N}(0, \rho^2 A_{\boldsymbol{\theta}} A_{\boldsymbol{\theta}}^\top)} [\mathcal{L}_D(\boldsymbol{\theta} + \boldsymbol{\epsilon})]$$

$$\leq \max_{\|\boldsymbol{\epsilon}\|_2^2 \leq \gamma^2, \boldsymbol{\epsilon} \in \mathcal{T}_{\boldsymbol{\theta}} \mathcal{M}} \mathcal{L}_S(\boldsymbol{\theta} + \boldsymbol{\epsilon}) + \sqrt{\frac{\frac{r^2(\sqrt{d} + \sqrt{\log n})^2}{2\gamma^2} + \log \frac{n}{\delta} + \mathcal{O}(d)}{2n - 2}}$$

Since $\gamma \propto \rho$, by rescaling we can conclude that:

$$\mathcal{L}_D(\boldsymbol{\theta}) \leq \max_{\boldsymbol{\theta}' \in \mathcal{B}_{\boldsymbol{\theta}}^a(\rho; \mathcal{T})} \mathcal{L}_S(\boldsymbol{\theta}') + \sqrt{\frac{\frac{r^2(\sqrt{d} + \sqrt{\log n})^2}{2\rho^2} + \log \frac{n}{\delta} + \mathcal{O}(d)}{2(n - 1)}}$$

$\square$

### A.1.5 PROOF OF SECOND INEQUALITY

(*The upper-bound on the manifold* $\mathcal{M}$): For any small $\rho > 0$ and $\delta \in [0; 1]$, with a high probability $1 - \delta$ over training set $\mathcal{S}$ generated from a distribution $\mathcal{D}$, the following holds:

$$\mathcal{L}_D(\boldsymbol{\theta}) \leq \max_{\boldsymbol{\theta}' \in \mathcal{B}_{\boldsymbol{\theta}}(\rho; \mathcal{M})} \mathcal{L}_S(\boldsymbol{\theta}') + \mathcal{O}\left( C(\mathcal{M})\rho^2 + \sqrt{\frac{\frac{d}{\rho^2} + d + \log \frac{n}{\delta}}{n - 1}} \right)$$

*Proof.* First, we recall the definition of the neighborhood on a manifold that $\mathcal{B}_{\boldsymbol{\theta}}(\rho, \mathcal{M}) = R_{\boldsymbol{\theta}}(\mathcal{B}_\rho^o(\boldsymbol{\theta}, \mathcal{T}))$. Indeed, for any $\varepsilon > 0$, there exists $\boldsymbol{\epsilon}_1, \boldsymbol{\epsilon}_2 \in \mathcal{T}_{\boldsymbol{\theta}} \mathcal{M}$ such that

$$\max_{\boldsymbol{\epsilon} \in \mathcal{T}_{\boldsymbol{\theta}} \mathcal{M}, \|\boldsymbol{\epsilon}\|_2^2 \leq \rho^2} \mathcal{L}_S(\boldsymbol{\theta} + \boldsymbol{\epsilon}) \geq \mathcal{L}_S(\boldsymbol{\theta} + \boldsymbol{\epsilon}_1) \geq \max_{\boldsymbol{\epsilon} \in \mathcal{T}_{\boldsymbol{\theta}} \mathcal{M}, \|\boldsymbol{\epsilon}\|_2^2 \leq \rho^2} \mathcal{L}_S(\boldsymbol{\theta} + \boldsymbol{\epsilon}) - \varepsilon$$

and also

$$\max_{\boldsymbol{\theta}' \in \mathcal{B}_{\boldsymbol{\theta}}(\rho; \mathcal{M})} \mathcal{L}_S(\boldsymbol{\theta}') = \max_{\boldsymbol{\epsilon} \in \mathcal{T}_{\boldsymbol{\theta}} \mathcal{M}, \|\boldsymbol{\epsilon}\|_2^2 \leq \rho^2} \mathcal{L}_S(R_{\boldsymbol{\theta}}(\boldsymbol{\epsilon})) \geq \mathcal{L}_S(R_{\boldsymbol{\theta}}(\boldsymbol{\epsilon}))$$

$$\geq \max_{\boldsymbol{\epsilon} \in \mathcal{T}_{\boldsymbol{\theta}} \mathcal{M}, \|\boldsymbol{\epsilon}\|_2^2 \leq \rho^2} \mathcal{L}_S(R_{\boldsymbol{\theta}}(\boldsymbol{\epsilon}_2)) - \varepsilon.$$

Combine the two inequalities together, we have the following:

$$\max_{\boldsymbol{\epsilon} \in \mathcal{T}_{\boldsymbol{\theta}} \mathcal{M}, \|\boldsymbol{\epsilon}\|_2^2 \leq \rho^2} \mathcal{L}_S(\boldsymbol{\theta} + \boldsymbol{\epsilon}) - \max_{\boldsymbol{\epsilon} \in \mathcal{T}_{\boldsymbol{\theta}} \mathcal{M}, \|\boldsymbol{\epsilon}\|_2^2 \leq \rho^2} \mathcal{L}_S(R_{\boldsymbol{\theta}}(\boldsymbol{\epsilon})) \leq \mathcal{L}_S(\boldsymbol{\theta} + \boldsymbol{\epsilon}_1) - \mathcal{L}_S(R_{\boldsymbol{\theta}}(\boldsymbol{\epsilon}_2)) + 2\varepsilon$$

in which $\boldsymbol{\epsilon}_1, \boldsymbol{\epsilon}_2 \in \mathcal{T}_{\boldsymbol{\theta}} \mathcal{M}$ such that $\|\boldsymbol{\epsilon}_1\|_2^2, \|\boldsymbol{\epsilon}_2\|_2^2 \leq \rho^2$.

Under the assumption that the model space is a compact manifold, it means that the domain of $\nabla \mathcal{L}_S$ is bounded. Therefore $\nabla \mathcal{L}_S$ is bounded by a constant $L$, which follows that

$$\max_{\boldsymbol{\epsilon} \in \mathcal{T}_{\boldsymbol{\theta}} \mathcal{M}, \|\boldsymbol{\epsilon}\|_2^2 \leq \rho^2} \mathcal{L}_S(\boldsymbol{\theta} + \boldsymbol{\epsilon}) - \max_{\boldsymbol{\epsilon} \in \mathcal{T}_{\boldsymbol{\theta}} \mathcal{M}, \|\boldsymbol{\epsilon}\|_2^2 \leq \rho^2} \mathcal{L}_S(R_{\boldsymbol{\theta}}(\boldsymbol{\epsilon})) \leq \mathcal{L}_S(\boldsymbol{\theta} + \boldsymbol{\epsilon}_1) - \mathcal{L}_S(R_{\boldsymbol{\theta}}(\boldsymbol{\epsilon}_2)) + 2\varepsilon$$

$$\leq \left( \mathcal{L}_S(\boldsymbol{\theta} + \boldsymbol{\epsilon}_1) - \mathcal{L}_S(\boldsymbol{\theta} + \boldsymbol{\epsilon}_2) \right) + \left( \mathcal{L}_S(\boldsymbol{\theta} + \boldsymbol{\epsilon}_2) - \mathcal{L}_S(R_{\boldsymbol{\theta}}(\boldsymbol{\epsilon}_2)) \right) + 2\varepsilon$$

Regarding the first term of the inequality above, we have

$$\mathcal{L}_S(\boldsymbol{\theta} + \boldsymbol{\epsilon}_1) - \mathcal{L}_S(\boldsymbol{\theta} + \boldsymbol{\epsilon}_2) = \nabla \mathcal{L}_S(C)(\boldsymbol{\epsilon}_1 - \boldsymbol{\epsilon}_2) \leq L\gamma$$

in which $C$ in the segment connect two points, since $\theta + \varepsilon_1$ is a point in high dimensional space. According to Lemma 1 in Boumal et al. (2018), we can bound the second term as:

$$\mathcal{L}_S(\boldsymbol{\theta} + \boldsymbol{\epsilon}_2) - \mathcal{L}_S(R_{\boldsymbol{\theta}}(\boldsymbol{\epsilon}_2)) \leq L\|R_{\boldsymbol{\theta}}(\boldsymbol{\epsilon}_2) - \boldsymbol{\theta} - \boldsymbol{\epsilon}_2\|_F \leq LC(\mathcal{M})\|\boldsymbol{\epsilon}_2\|_2^2 \leq LC(\mathcal{M})\|\gamma\|_2^2.$$

It follows that

$$
\begin{aligned}
\mathcal{L}_{\mathcal{D}}(\boldsymbol{\theta}) \leq{} & \max_{\boldsymbol{\epsilon} \in \mathcal{T}_{\boldsymbol{\theta}}\mathcal{M}, \|\boldsymbol{\epsilon}\|_2^2 \leq \rho^2} \mathcal{L}_S(\boldsymbol{\theta}+\boldsymbol{\epsilon}) + \sqrt{\frac{\mathcal{O}(\frac{d}{\rho^2}+d+\log\frac{n}{\delta})}{n-1}} \\
\leq{} & \max_{\boldsymbol{\epsilon} \in \mathcal{T}_{\boldsymbol{\theta}}\mathcal{M}, \|\boldsymbol{\epsilon}\|_2^2 \leq \rho^2} \mathcal{L}_{\mathcal{S}}(R_{\boldsymbol{\theta}}(\boldsymbol{\epsilon})) \\
& + \left[\max_{\boldsymbol{\epsilon} \in \mathcal{T}_{\boldsymbol{\theta}}\mathcal{M}, \|\boldsymbol{\epsilon}\|_2^2 \leq \rho^2} \mathcal{L}_{\mathcal{S}}(\boldsymbol{\theta}+\boldsymbol{\epsilon}) - \max_{\boldsymbol{\epsilon} \in \mathcal{T}_{\boldsymbol{\theta}}\mathcal{M}, \|\boldsymbol{\epsilon}\|_2^2 \leq \rho^2} \mathcal{L}_{\mathcal{S}}(R_{\boldsymbol{\theta}}(\boldsymbol{\epsilon}))\right] + \sqrt{\frac{\mathcal{O}(\frac{d}{\rho^2}+d+\log\frac{n}{\delta})}{n-1}} \\
\leq{} & \max_{\boldsymbol{\theta}' \in \mathcal{B}_{\boldsymbol{\theta}}(\rho;\mathcal{M})} \mathcal{L}_{\mathcal{S}}(\boldsymbol{\theta}') + \sqrt{\frac{\mathcal{O}(\frac{d}{\rho^2}+d+\log\frac{n}{\delta})}{n-1}} + \varepsilon \\
& + \left[\mathcal{L}_S(\boldsymbol{\theta}+\boldsymbol{\epsilon}_1) - \mathcal{L}_S(\boldsymbol{\theta}+\boldsymbol{\epsilon}_2)\right] + \left[\mathcal{L}_S(\boldsymbol{\theta}+\boldsymbol{\epsilon}_2) - \mathcal{L}_S(R_{\boldsymbol{\theta}}(\boldsymbol{\epsilon}_2))\right] + \sqrt{\frac{\mathcal{O}(\frac{d}{\rho^2}+d+\log\frac{n}{\delta})}{n-1}} \\
\leq{} & \max_{\boldsymbol{\theta}' \in \mathcal{B}_{\boldsymbol{\theta}}(\rho;\mathcal{M})} \mathcal{L}_{\mathcal{S}}(\boldsymbol{\theta}') + LC(\mathcal{M})\rho^2 + L\rho + \sqrt{\frac{\mathcal{O}(\frac{d}{\rho^2}+d+\log\frac{n}{\delta})}{n-1}} + 2\varepsilon \\
\leq{} & \max_{\boldsymbol{\theta}' \in \mathcal{B}_{\boldsymbol{\theta}}(\rho;\mathcal{M})} \mathcal{L}_{\mathcal{S}}(\boldsymbol{\theta}') + \mathcal{O}\left(C(\mathcal{M})\gamma^2 + \sqrt{\frac{\frac{d}{\rho^2}+d+\log\frac{n}{\delta}}{n-1}}\right)
\end{aligned}
$$

which concludes our proof. $\qquad\square$

**Lemma 1.** *(Boumal et al., 2018) There exists a constant $C(\mathcal{M}) > 0$ such that for any $\boldsymbol{\theta} \in \mathcal{M}$ and $\boldsymbol{\epsilon} \in \mathcal{T}_{\boldsymbol{\theta}}\mathcal{M}$, the following holds:*

$$\|R_{\boldsymbol{\theta}}(\boldsymbol{\epsilon}) - \boldsymbol{\theta} - \boldsymbol{\epsilon}\|_F \leq C(\mathcal{M})\|\boldsymbol{\epsilon}\|_F^2$$

*The constant $C(\mathcal{M})$ value depends on the manifold structure and may scale with dimensions for general manifolds. However, for retractions on the Stiefel manifold, the constant is independent of $(d, k)$ and can be computed explicitly. Specifically, when using the QR factorization or the polar decomposition as the retraction, we have $C(\mathcal{M}) = 1 + \sqrt{2}/2$.*

## A.2 ADDITIONAL EXPERIMENTS

### A.2.1 PER-EPOCH RUNTIME

This ablation compares the single-epoch wallclock runtimes of SGD, SAM, and RSAM. Indeed, it is expected that SAM and RSAM take at least twice as long as SGD because both SAM and RSAM involve double backward-forward each iteration. We especially note that since RSAM involves additional computations on a manifold, it is expected that RSAM would take longer than SAM. As shown in Table 5, while RSAM improves the final performance, its runtime is only about 6% slower than SAM, therefore worth the tradeoff.

| Method | CIFAR100 | | CIFAR10 | | AirCraft | |
|---|---|---|---|---|---|---|
| | ResNet34 | ResNet50 | ResNet34 | ResNet50 | ResNet34 | ResNet50 |
| SGD | $21.5_{\pm1.73}$ | $40.1_{\pm2.96}$ | $21.4_{\pm1.72}$ | $38.8_{\pm3.05}$ | $57.6_{\pm2.59}$ | $114.5_{\pm4.77}$ |
| SAM | $49.1_{\pm1.68}$ | $83.9_{\pm2.79}$ | $48.8_{\pm1.62}$ | $84.7_{\pm2.68}$ | $125.3_{\pm1.3}$ | $245.6_{\pm4.30}$ |
| RSAM | $52.6_{\pm1.66}$ | $88.7_{\pm3.14}$ | $51.1_{\pm1.90}$ | $88.3_{\pm2.79}$ | $133.0_{\pm1.4}$ | $259.2_{\pm3.82}$ |

**Table 5:** Per-epoch wall-clock runtime in seconds.

### A.2.2 SHARPNESS EVOLUTION AND HESSIAN SPECTRAL

Throughout this work, we have designed RSAM to actively seek local minima in the regions within a manifold with both low loss value and low sharpness. In this section, to further verify whether

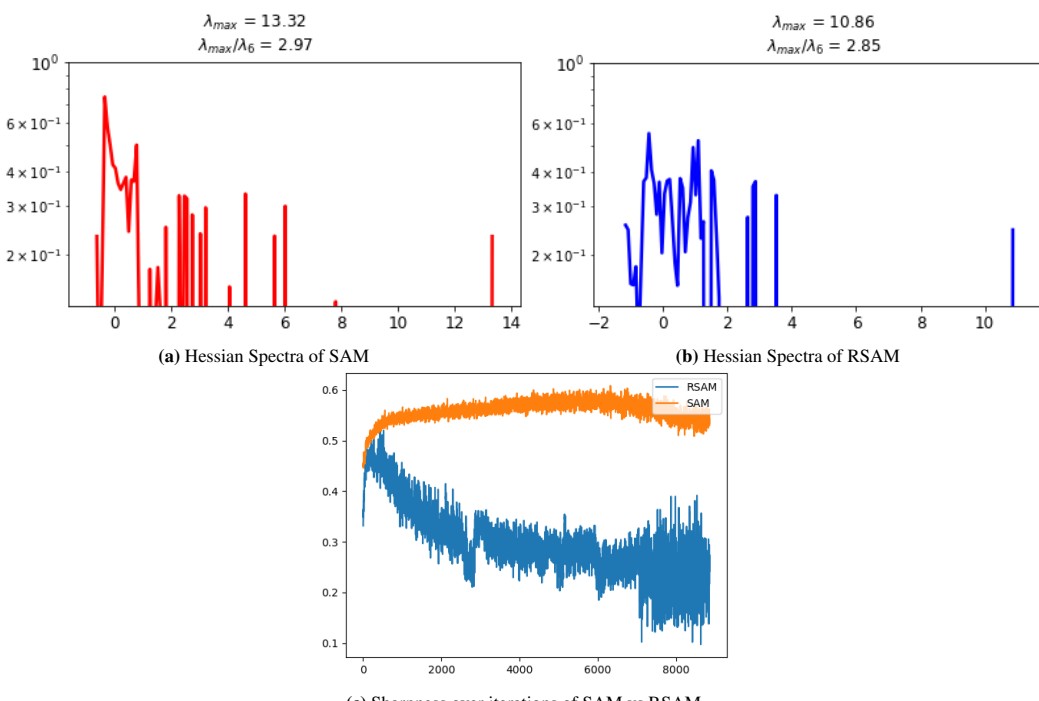

**(a)** Hessian Spectra of SAM          **(b)** Hessian Spectra of RSAM

**(c)** Sharpness over iterations of SAM vs RSAM.

**Figure 4:** The spectrum of the Hessian at the termination of the training phase with SAM vs. RSAM (above) and the evolution of sharpness over iterations (below). The results are reported on the CIFAR100 dataset with the SupCon loss function. $\rho$ in both methods is set to $0.1$

RSAM found the low-sharpness region, we first contrast the spectral of the Hessian for ResNet34 trained on CIFAR100 for 400 steps with RSAM and SAM. Indeed, the model trained with RSAM has a lower maximum eigenvalue (10.86 of RSAM vs. 13.32 of SAM), and RSAM has a flatter eigenvalue distribution as shown in the Figure 4, therefore suggests that RSAM entered the lower-sharpness region on the loss landscape. Besides, we also report the sharpness evolutions over iterations as shown in Figure 4c. These results together indicate that RSAM successfully seeks points in lower-sharpness regions within the loss landscape.