# OpenReview forum: "RSAM: Learning on Manifolds with Riemannian Sharpness-Aware Minimization"
_ICLR.cc/2024/Conference — ICLR 2024 Conference Withdrawn Submission_

### Official Review · Reviewer_gTdN · 2023-10-31

**Soundness:** 3 good
**Presentation:** 1 poor
**Contribution:** 2 fair
**Rating:** 5
**Confidence:** 3

**Summary:**

This work introduces the novel notion of Sharpness on Riemannian manifolds and proposes a tighter upper bound. The authors also introduce RSAM, which considers the parameter space’s intrinsic geometry and seeks regions with flat surfaces on Riemannian manifolds. They also provide empirical results to show the effectiveness of RSAM.

**Strengths:**

The paper is the first to consider the sharpness of parameters lying on a manifold, which has potential to be an interesting branch of SAM. The empirical results are supportive and reasonable.

**Weaknesses:**

1. There is no equation number in section 3.3. Also, what is the RHS of the second equation in section 3.3?
2. The code link provided has expired, so I can't reproduce your results.
3. The error bars in Table. 1 and 2 are missing.

**Questions:**

1. How did you get eq 2 from the last row of the eq above eq 2? Why omit $\mathcal{L}_{\mathcal{S}}$?  (please add equation numbers)
2. Could you explain how $\mathcal{R}_\theta$ works in practice? Is it like a projection to manifold space?
3. Why in the last line of Alg 1 there is another $\mathcal{R}_{\theta_t}$? Could you explain?
4. Just curious, what if $\mathbf{D}$ has more degree of freedom, like a function of all $\theta$?

---

> ### Author Response · Authors · 2023-11-22
>
> Hi, reviewer gTdN. Thank you for your comments. We would like response as follows:
> + Firstly, it was a typo in page 4 that $\mathcal{L}_\mathcal{S}$ was omitted, we have fixed the issue.
>
> + The second equation has no RHS, it essentially means that motivated by the theorems above, we should instead solve the optimization problem $min\_{\theta \in \mathcal{M}} \mathcal{L}^{RSAM}\_\mathcal{S}(\theta)$ instead of $min\_{\theta \in \mathcal{M}} \mathcal{L}\_{\mathcal{D}}(\theta)$ as mentioned at the beginning of Section (3.2).
>
> + The operation $\mathcal{R}\_\theta$ is the well-known retraction operation on a manifold. Specifically, it will be a smooth mapping from the tangent space to the manifold: $\mathcal{R}\_{\theta}: \mathcal{T}\_{\theta} \mathcal{M} \to \mathcal{M}$. This retraction map satisfies $\mathcal{R}\_\theta(0\_\theta) = \theta$,
> where $0\_\theta$ denotes the zero element of the vector space $\mathcal{T}\_\theta \mathcal{M}$, and $\frac{d}{dt}\_{t = 0} \mathcal{R}\_{\theta}(t \xi) = \mathcal{T}\_{0\_\theta} \mathcal{R}\_{\theta}(\xi) = \xi$ for all $\xi \in \mathcal{T}\_{\theta}\mathcal{M}$. In practice, a manifold can have multiple different retraction operations. For Stiefel manifold, we specifically used the operation mentioned in Section (4). In the last line of Alg 1, the old $R\_{\theta_t}$ denotes the same retraction operation at $\theta_t$, which is $\mathcal{R}\_{\theta_t}$. However, it was a typo that $R$ is not capitalized, we have also fixed that in the revision.
>
> + We believe that $D$ having more degree of freedom can also be a good idea to explore because it potentially can capture more information within the loop. In our experiments, we chose $D$ to be a simple matrix based on a single $\theta$ for the sake of computational tradefoff, and we demonstrate that it can improve the final result with only a little tradeoff. Last but not least, we have also included the proper link to our code. Thank you for your feedbacks.

---

> > ### Comment · Reviewer_gTdN · 2023-12-04
> >
> > I have checked the responses as well as comments from other reviewers. The rebuttal does solve some of my questions, but I still think the paper will benefit from more rigorous and careful writing. I will keep my score towards rejection.

---

### Official Review · Reviewer_efpX · 2023-10-31

**Soundness:** 1 poor
**Presentation:** 1 poor
**Contribution:** 1 poor
**Rating:** 3
**Confidence:** 5

**Summary:**

This paper proposes employment of SAM on Riemannian manifolds. The proposed methods were explored on several image classification datasets with Resnet architectures.

**Strengths:**

The proposed RSAM boosts accuracy of SAM in a few image classification tasks.

**Weaknesses:**

There are two major problems with the paper.

First, several statements used to describe the proposed method and its implementation in the paper are not clear.

Second, the experimental analyses are limited. The proposed RSAM should be examined on additional DNN architectures, datasets and larger category of Riemannian manifolds in comparison with the other Riemannian optimizers and SAM optimizers.

**Questions:**

-	In the paper, it is stated that “we imposed orthogonality on a single convolutional layer in the middle of the architecture in all settings”. This statement is not clear. How did you define the “single convolution layer” more precisely? Did you just add orthogonality to one layer?

-	It is stated that “Since U is constrained to lies on the Stiefel manifold, we will optimize it with RSAM, and the rest of the parameters, including the backbone and the diagonal matrix S, will be learned via traditional optimizers such as SAM or SGD”. The S can be optimized using RSAM as well, since it is a diagonal matrix residing on a Riemannian manifold. How does the accuracy change when it is optimized by SAM, RSAM, SGD?

-	Can you provide the results obtained using additional optimizers such as Riemannian SGD, Adam, Riemannian Adam, and AdamW?

-	A similar work was recently published in the Neurips; Yun and Yang, Riemannian SAM: Sharpness-Aware Minimization on Riemannian Manifolds. A direct comparison with this work may not be possible since their code/paper is not completely available. However, as they mentioned in the abstract, such a work on SAM should be compared with the other SAM methods such as Fisher SAM on a more general category of Riemannian manifolds.

---

> ### Author Response · Authors · 2023-11-22
>
> Hi, reviewer efpX. Thank you for your comments. There are a few things we want to clarify and we would like to respond to your comments as follows:
> + Firstly, about the phrase “we imposed orthogonality on a single convolutional layer in the middle of the architecture in all settings”, what we mean in this statement is that we impose orthogonality on a convolutional layer, which is different from imposing orthogonality on a single kernel. Specifically, with PyTorch, on ResNet34 we can extract this layer as **model.layer1.1.conv1.weight**. We found that only imposing orthogonality on this single layer improves the performance notably, while we also have the freedom to put the constraint on multiple layers.
>
>  + The matrix $S$ can also be optimized with SAM and SGD. In our experimental results, the "SAM" row means that the whole architecture, including the matrix $S$, is optimized with SAM. We need to clarify that in the decomposition in section (4.1), the matrix $S$ is not constrained to reside on the Stiefel manifolds, so it has the freedom to have any values.
>
> + We will take into account your third comment and provide additional results on those baselines. Among those baselines, we have experimented with Riemannian SGD. Specifically, we optimized the matrix $U$ with RSGD and other parameters with SGD. What we found was that the performance was about the same as SAM in average across the settings on average.
>
> + Also, thank you for informing us about the work "Riemannian SAM: Sharpness-Aware Minimization on Riemannian Manifolds". It was a concurrent work that we were not informed of while working on this direction. Nevertheless, our approach has slightly different mathematical development, theoretical results, and application directions. Indeed, as you mentioned, it is also desirable to include comparisons with FisherSAM, Riemannian Adam, and AdamW. We will take that into account.

---

### Official Review · Reviewer_damX · 2023-11-06

**Soundness:** 1 poor
**Presentation:** 1 poor
**Contribution:** 2 fair
**Rating:** 3
**Confidence:** 4

**Summary:**

This paper proposes Riemannian sharpness-aware minimization (RSAM), which extends the original SAM algorithm to the case of parameters residing within Riemannian manifolds. The authors first establish the notion of sharpness for loss landscapes defined on Riemannian manifolds. Subsequently, they provide a theoretical analysis relating the sharpness to the generalization gap and propose the RSAM algorithm, designed to minimize the sharpness augmented loss. To demonstrate the effectiveness of the RSAM algorithm, experiments on image classification and constrastive learning tasks are performed, focusing on the parameters defined on Stiefel manifolds.

**Strengths:**

- This paper provides an efficient extension of the SAM algorithm for constrained parameter spaces, accompanied by a theoretical analysis of the generalization gap.
- Experimental results indicate some performance improvements.

**Weaknesses:**

- There seems to be an inconsistency in defining neighborhoods in Section 3.1 and the choice of Riemannian metric in the experiments, which can confuse the readers significantly. The Riemannian metric $D_\theta$ seems to be the ambient space metric. If this is the case, the norm $||\cdot||$ should be defined using $D_\theta$, but all derivations in Section 3 are based on assuming $D_\theta = I$ (as per the proofs in Appendix A.1), implying the Euclidean ambient space. However, experiments employ $D_\theta$ different from the identity, of which the choice seems arbitrary.
- The claim of providing a tighter bound than SAM should be more carefully nuanced. The parameter spaces possessing a manifold structure are of little concern in the original SAM paper. Therefore, it would be more accurate to state that the provided bound is tighter ‘when the parameter spaces have much smaller dimensionality than the ambient space’ rather than making a general comparison to SAM.
- RSAM seems to be a straightforward generalization of SAM to Riemannian manifolds, which might be considered a minor contribution unless the paper includes case studies applying the proposed algorithm to a range of Riemannian manifolds. While the Stiefel manifold considered in the paper is a relevant example, including application examples on other Riemannian manifolds would be beneficial.
- Even though the experimental results suggest some performance advantages of using RSAM, the analysis is not sufficiently thorough. The primary reason for the improvement appears to be the use of the R-Stiefel layer, and the comparison of RSAM with SGD and SAM without the R-Stiefel layer may not be fair. For a more precise analysis of the generalization benefit of RSAM, further experimental studies are needed, such as comparing it to Riemannian SGD, which also employs the R-Stiefel layer.
- The paper would benefit from clearer writing, particularly in Section 4.

**Questions:**

- How do the choices of hyperparameters, such as $\rho$ for RSAM and SAM, influence the results in Section 5, and how were these hyperparameters selected?
- When obtaining the Hessian spectral in Appendix A.2.2, shouldn’t the geometry, e.g., Riemannian metric $D_\theta$, be considered?
- The concept of retraction is used frequently without a precise definition. How is retraction defined?

[Typos]
- At the beginning of Section 3.2, it should read: $\mathcal{M} \subseteq \mathbb{R}^k$.
- In Section 3.3, the omission of $\mathcal{L}_\mathcal{S}$ in deriving the objective function and in equation (3) should be corrected.

---

### Official Review · Reviewer_dxfe · 2023-11-12

**Soundness:** 2 fair
**Presentation:** 2 fair
**Contribution:** 2 fair
**Rating:** 5
**Confidence:** 3

**Summary:**

This paper extends the Sharpness-Aware Minimization (SAM) approach to the Riemannian manifolds, e.g. when the learned models should satisfy certain constraints. Theoretically, the paper demonstrates that the generalization gap on manifolds scales with $\mathcal{O}(\sqrt{d})$ where $d$ is the dimension of the manifold and could be much smaller than $k$, that is the dimension of the ambient space. The paper provides experimental evaluations and compares the proposed method RSAM with other benchmarks such as SAM and SGD for supervised and self-supervised learning tasks.

**Strengths:**

I think the motivation and the idea of RSAM are valid and interesting. The result of Theorem 1 in which the $\mathcal{O}(\sqrt{k})$ factor in SAM's generalization gap reduces to $\mathcal{O}(\sqrt{d})$ on manifold seems quite interesting.

**Weaknesses:**

The paper is fairly difficult to follow in some parts. I suggest to elaborate more on the prior work on "learning on manifolds" and its technical literature. For instance, it seems that the proof of Theorem 1 relies substancially on results from (Boumal et al., 2018) and Lemma which are only touched on without sufficient discussion. Moreover, I could find several typos in math and inexact statements thoughout the paper. Pleasse my comments below.

**Questions:**

- Proof of Theorem 1 states that "Since the loss function L is $K$-Lipschitz, we have..." while the Lipschitz assumption is not mentioned in the theorem's statement or elsewhere. Could the authors clarify this?
- In proof of Theorem 1, what does $\tilde{{\theta}}$ denote? And what is a "logarithm map"?
- What is $v_{\theta}$ in Proposition 1? I assume it should be $u_{\theta}$?
- Section 3.1 would be easier to follow if the authors could add more elaboration on the retraction operator $R_{\theta}$ before going to Section 3.2.
- In experiments, the $\rho$ parameters for SAM and RSAM are different. I wonder if this is a fair comparison given that now the geometry of the manifold determines the robustness of RSAM as well. Could the authors elaborate on the effect of $\rho$ on the accuracies?
- What is the retraction operator considered in Lemma 1?

Minor comments:

- In Section 3.3, the second and third equations seem to miss $\mathcal{L}$ in the maximization objective.
- Equation (2) seems to be missing $\mathcal{L}_S(\theta)$ in the objective (compared to the previous derivation before eq. (2)).

---

> ### Author Response · Authors · 2023-11-23
>
> Hi reviewer dxfe, thank you for your comments. We would like to respond to your comments as follows:
>
> + In Proposition 1, $\textbf{v}_\vtheta^{\top} = \gradtheta\gL(\vtheta)^{\top}\textbf{D}_\vtheta$ as we mentioned at the beginning. However, we only used that in the proof for simplicity and is not needed for the proposition statement, so we can safely removed that. We have updated that in the revision.
> .
> + In the proof of Theorem 1, $\tilde{\theta}$ is the image of $\theta$ under the logarithm map. The logarithmic map is a fundamental concept in Riemannian manifolds. Specifically, Let $v \in \mathcal{T}\_\mathcal{\theta} \mathcal{M}$ be a tangent vector to the manifold at $\theta$. Then there is a unique geodesic $\gamma:[0,1] → \mathcal{M}$ satisfying $\gamma(0) = \theta$ with initial tangent vector $\gamma'(0) = v$. The corresponding exponential map is defined by $\exp\_{\theta}(v) = \gamma(1)$. Hence, the logarithmic map is the inverse of this exponential map.
>
> + Regarding the effect of $\rho$ on the accuracy, we will take into account your comment for further improvement. Currently, from what we have tried so far, $\rho$ for RSAM is chosen similarly to that of SAM in the original SAM work, and the accuracies in those sensible values are not differed significantly.
>
> + The loss function $\mathcal{L}$ should be assumed to be Lipschitz, we have included that in the revision and also the typos that you mentioned in the minor comments.